# Improving Deep Learning Optimization through Constrained Parameter Regularization

**Jörg K.H. Franke**
University of Freiburg, Germany

**Michael Hefenbrock**
RevoAI, Karlsruhe, Germany

**Gregor Koehler**
German Cancer Research Center (DKFZ)
Heidelberg, Germany

**Frank Hutter**
ELLIS Institute Tübingen, Germany
University of Freiburg, Germany

## Abstract

Regularization is a critical component in deep learning. The most commonly used approach, weight decay, applies a constant penalty coefficient uniformly across all parameters. This may be overly restrictive for some parameters, while insufficient for others. To address this, we present Constrained Parameter Regularization (CPR) as an alternative to traditional weight decay. Unlike the uniform application of a single penalty, CPR enforces an upper bound on a statistical measure, such as the $L_2$-norm, of individual parameter matrices. Consequently, learning becomes a constraint optimization problem, which we tackle using an adaptation of the augmented Lagrangian method. CPR introduces only a minor runtime overhead and only requires setting an upper bound. We propose simple yet efficient mechanisms for initializing this bound, making CPR rely on no hyperparameter or one, akin to weight decay. Our empirical studies on computer vision and language modeling tasks demonstrate CPR's effectiveness. The results show that CPR can outperform traditional weight decay and increase performance in pre-training and fine-tuning.

## 1 Introduction

Deep neural networks are the bedrock of many state-of-the-art machine learning applications [1]. While these models have exhibited unparalleled expressivity, they also possess millions, sometimes trillions, of parameters [2]. This massive capacity makes them susceptible to overfitting, where models memorize nuances of the training data but underperform on unseen examples. To mitigate this, many different regularization techniques have been adopted, with weight decay and $L_2$ regularization [3, 4, 5] being the most popular. $L_2$ regularization penalizes the squared magnitude of model parameters and (decoupled) weight decay (which is equivalent to $L_2$ regularization for non-adaptive gradient algorithms [6]) multiplies all weights with a constant at every step. This seemingly simple act offers numerous benefits by curbing the growth of individual weights, reducing the risk of relying on any particular feature excessively, and thus promoting model generalization.

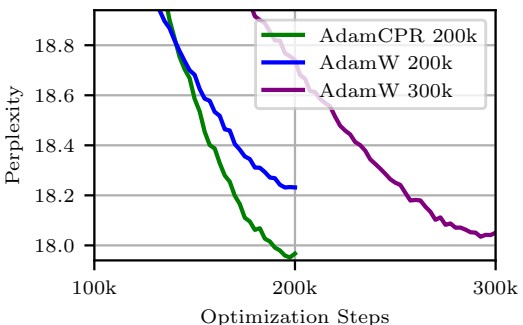

Figure 1: GPT2s training using Adam with weight decay or CPR (`Kappa-IP`). AdamCPR outperforms AdamW with the same budget and only requires 2/3 of the budget to reach the same score.

38th Conference on Neural Information Processing Systems (NeurIPS 2024).

However, not all parameters in a neural network have the same role or importance and different weights could benefit from different regularization. Similarly, it is unclear if a single weight decay value is optimal for the entire duration of optimization, especially for large-scale training. Indeed, Ishii and Sato [7] showed that a small deep learning model could benefit from layer-wise weight decay values, and various works showed that scheduling weight decay could improve final performance [8, 9, 10, 11]. This indicates that a dynamic penalty for each individual parameter matrix (e.g., a weight matrix in a linear layer) could be beneficial for neural network training. Since both scheduling and parameter-wise weight decay require additional hyperparameters that are often sensitive to the task, we propose a different approach to obtain customized, dynamic parameter regularization. Instead of uniformly penalizing weights, we propose to keep them in a certain range, thus ensuring stability without imposing regularization where it is unnecessary. Constraining parameters, especially based on statistical measures like the $L_2$ norm, provide a flexible and adaptive form of regularization that accounts for the heterogeneity of parameters.

In this paper, we propose *Constrained Parameter Regularization (CPR)*, which enforces an upper bound on a statistical measure of individual parameter matrices. Consequently, regularization is expressed as a constrained optimization problem, which we address by an adaptation of the augmented Lagrangian method. The regularization of each parameter matrix is handled by a separate constraint and Lagrange multiplier, resulting in an individual regularization strength that adapts over time. The method requires the selection of desired constraint values as well as an update rate for the Lagrange multipliers. We found that the update rate can be fixed to 1.0. For choosing constraint values, we introduce four strategies, three of which require a single hyperparameter, while the last one is hyperparameter-free. We show in our experiments performance improvements over weight decay when pre-training or finetuning models for image classification (CIFAR100 and ImageNet), language modeling (OpenWebText), and medical image segmentation. For example, when training a GPT2s model, we achieved the same performance as AdamW but only require $2/3$ of the budget, see Figure 1. Applying our method for fine-tuning, we find performance improvements and less catastrophic forgetting. In the following, and after discussing related work (Section 2) and background on weight decay and the augmented Lagrangian method (Section 3), we make the following contributions:

- Introducing CPR for individualized and dynamic weight regularization[1]. Specifically, formulating regularization as a constraint optimization problem and proposing CPR as a solution (Section 4.1).
- Identifying four different strategies for initializing this constraint (Section 4.3). One of them, `Kappa-WS`, has a strong default that outperforms tuned AdamW; and another one, `Kappa-IP`, is entirely hyperparameter-free and yields even better performance in pre-training.
- Showing improved performance over weight decay in image classification, medical image segmentation, and pretraining and fine-tuning language models (Section 5).

## 2   Related Work

Weight decay is an effective regularization technique to improve the generalization and model performance [12], and the idea of adapting parameter regularization during training is not new. Lewkowycz and Gur-Ari [8] investigated the effect of $L_2$ regularization on overparameterized networks and found the time it takes the network to reach peak performance is proportional to the $L_2$ regularization parameter. They proposed an initialization scheme for $L_2$ regularization and an annealing schedule for the $L_2$ parameter. Yun et al. [9] use a combination of weight decay scheduling and knowledge distillation to improve performance on computer vision tasks. More recent works on self-supervised vision transformers also use a weight decay schedule [10, 11]. In contrast to our work, none of these proposes a dynamic and individual adaptation of each regularized parameter matrix. Also, a schedule comes with varying hyperparameter choices while CPR adapts arbitrarily many parameter matrices with only two hyperparameters (out of which one is fixed in all our experiments). Instead of using a schedule, Nakamura and Hong [13] proposes *AdaDecay*, where the $L_2$ penalty is scaled by standardized gradient norms and a sigmoid function. Ghiasi et al. [14] propose another gradient-based approach, *Adaptive Weight Decay (AWD)*, which dynamically adjusts the weight decay based on the ratio of weight norms to gradient norms to balance the contributions from the cross-entropy and regularization losses aiming to improve the robustness. AMOS [15]

---

[1]Please find our implementation under `https://github.com/automl/CPR`.

leverages model-specific information for initialization and gradients to adapt L2 regularization during the training. Another way to regularize parameters is to fix the norm of individual parameter matrices [16], to schedule the weight norm [17], or to limit the total norm of all parameters [18] to a fixed value. This fixed value is a more sensitive hyperparameter than the hyperparameter in our work.

Our proposed method is not the first to use Lagrangian methods in machine learning [19]. Its application in deep learning so far focuses on variational methods and generative models: Rezende and Viola [20] introduced the *Generalized ELBO with Constrained Optimization* algorithm to optimize VAEs using Lagrange multipliers optimized by the min-max scheme, and Kohl et al. [21] and Franke et al. [22] adapted the Lagrangian method from Rezende and Viola [20] to train probabilistic U-nets and probabilistic Transformer models. While these works adopt Lagrangian methods to handle several losses in joint optimization problems, our work leverages them to enable individual regularization strengths.

## 3 Background

### 3.1 $L_2$ Regularization and Weight Decay

Regularization methods, such as $L_2$-regularization or weight decay, are commonly used to restrict parameter updates and enhance generalization by reducing unnecessary complexity [3, 4, 5]. Both can be motivated by introducing a "cost" to weight magnitudes. Specifically, in $L_2$-regularization, instead of minimizing only the loss function $L(\boldsymbol{\theta}, \boldsymbol{X}, \boldsymbol{y})$ with parameters $\boldsymbol{\theta}$ and data $\mathcal{D} = \{(\boldsymbol{X}_n, \boldsymbol{y}_n)\}_{n=0}^{N}$, a weighted penalty (regularization) term $R(\boldsymbol{\theta})$ is added to the loss, resulting in the training objective

$$\min_{\boldsymbol{\theta}} \quad L(\boldsymbol{\theta}, \boldsymbol{X}, \boldsymbol{y}) + \gamma \cdot R(\boldsymbol{\theta}),$$

where $R(\boldsymbol{\theta}) = \frac{1}{2}\|\boldsymbol{\theta}\|_2^2$ denotes the regularization function and $\gamma \in \mathbb{R}^+$ the strength of the penalty. On the other hand, weight decay directly modifies the update rule of the parameters to

$$\boldsymbol{\theta}_{t+1} \leftarrow \boldsymbol{\theta}_t + \mathrm{Opt}(L, \eta) - \eta \cdot \gamma \cdot \boldsymbol{\theta}_t,$$

where $\mathrm{Opt}(L, \eta)$ denotes an optimizer providing the gradient-based update at iteration $t$ and $L = L(\boldsymbol{\theta}_t, \boldsymbol{X}_t, \boldsymbol{y}_t)$ the loss. For example, $\mathrm{Opt}(L, \eta) = -\eta \cdot \nabla_{\boldsymbol{\theta}} L(\boldsymbol{\theta}_t, \boldsymbol{X}_t, \boldsymbol{y}_t)$ with learning rate $\eta \in \mathbb{R}^+$ in case of gradient descent. Thus, the main difference between weight decay and $L_2$-regularization is that the gradients of the regularization accumulate in momentum terms in the case of $L_2$-regularisation, while they are treated separately in (decoupled) weight decay. This has also been extensively discussed by Loshchilov and Hutter [6] with the introduction of the AdamW optimizer.

### 3.2 The augmented Lagrangian method

We briefly review the augmented Lagrangian method for constrained optimization, see e.g. Bertsekas [23], which our method is based on. For the derivation, we follow the motivation of Nocedal and Wright [24, pp. 523-524]. Consider the following inequality-constrained optimization problem

$$\underset{\boldsymbol{x}}{\text{minimize}} \ f(\boldsymbol{x}) \quad \text{s.t.} \quad c(\boldsymbol{x}) \leq 0,$$

with $f(\boldsymbol{x}) : \mathbb{R}^n \to \mathbb{R}$ and a constraint $c(\boldsymbol{x}) : \mathbb{R}^n \to \mathbb{R}$. One way to address the constraint is to find an equivalent, unconstrained problem with the same optimal solution. For example,

$$\underset{\boldsymbol{x}}{\text{minimize}} \ F(\boldsymbol{x}) \quad \text{with} \quad F(\boldsymbol{x}) = \max_{\lambda \geq 0} \ f(\boldsymbol{x}) + \lambda \cdot c(\boldsymbol{x}). \tag{1}$$

Unfortunately, even if $f(\boldsymbol{x})$ and $c(\boldsymbol{x})$ are differentiable, $F(\boldsymbol{x})$ is not differentiable. This is due to the maximization over $\lambda$ in $F(\boldsymbol{x})$, where in case of $c(\boldsymbol{x}) > 0$, $F(\boldsymbol{x}) \to \infty$. Consequently, we cannot run gradient-based optimization on this objective.

To alleviate this problem, we consider a smooth approximation of $F(\boldsymbol{x})$, namely

$$\hat{F}(\boldsymbol{x}, \lambda_t, \mu) = \max_{\lambda \geq 0} \ f(\boldsymbol{x}) + \lambda \cdot c(\boldsymbol{x}) - \frac{1}{2\mu}(\lambda - \lambda_t)^2, \tag{2}$$

where $\lambda_t \in \mathbb{R}$ may be seen as a point we wish to remain proximal to and $\mu \in \mathbb{R}^+$ as a factor determining the strength with which this proximity is enforced. For $\mu \to \infty$, $\hat{F}(\boldsymbol{x}, \lambda_t, \mu) \to F(\boldsymbol{x})$.

The maximization in $\hat{F}(\boldsymbol{x}, \lambda_t, \mu)$ has a closed form solution with $\lambda^\star = (\lambda_t + \mu \cdot c(\boldsymbol{x}))^+$, where $(\cdot)^+ = \max\{0, \cdot\}$, see Appendix A for the derivation. Consequently, $\hat{F}(\boldsymbol{x}, \lambda_t, \mu)$ can be written as

$$\hat{F}(\boldsymbol{x}, \lambda_t, \mu) = f(\boldsymbol{x}) + h(\boldsymbol{x}, \lambda_t, \mu)$$

with

$$h(\boldsymbol{x}, \lambda_t, \mu) = \begin{cases} c(\boldsymbol{x})(\lambda_t + \frac{\mu}{2}c(\boldsymbol{x})), & \text{if} \quad \lambda_t + \mu \cdot c(\boldsymbol{x}) \geq 0 \\ -\frac{1}{2\mu}\lambda_t^2 & \text{else.} \end{cases}$$

The constraint thus only interferes with the minimization (gradient) of $f(\boldsymbol{x})$ if $\lambda_t + \mu \cdot c(\boldsymbol{x}) \geq 0$. We can now try to solve the unconstrained problem $\underset{\boldsymbol{x}}{\text{minimize}}\ \hat{F}(\boldsymbol{x}, \lambda_t, \mu)$ with familiar methods, such as gradient descent, and obtain an approximate solution to the original problem. Specifically, the gradient of $\hat{F}(\boldsymbol{x}, \lambda_t, \mu)$ with respect to $\boldsymbol{x}$ is given by

$$\nabla_{\boldsymbol{x}}\hat{F}(\boldsymbol{x}, \lambda_t, \mu) = \nabla_{\boldsymbol{x}}f(\boldsymbol{x}) + \lambda^\star \cdot \nabla_{\boldsymbol{x}}c(\boldsymbol{x}).$$

The quality of the approximation, and thus the solution, clearly depends on $\mu$ and $\lambda_t$. To improve this approximation we can refine the choice of $\lambda_t$ via an iterative procedure and repeat the optimization with $\lambda_{t+1} \leftarrow \lambda^\star = (\lambda_t + \mu \cdot c(\boldsymbol{x}))^+$. Intuitively, if the previous minimization of $\hat{F}(\boldsymbol{x}, \lambda_t, \mu)$ resulted in an infeasible solution with $c(\boldsymbol{x}) > 0$, $\lambda_{t+1} > \lambda_t$. Hence, the minimization of $\hat{F}(\boldsymbol{x}, \lambda_{t+1}, \mu)$ likely results in a solution with less constraint violation. On the other hand, if $c(\boldsymbol{x}) \leq 0$, $\lambda_{t+1} \leq \lambda_t$. Subsequently, the influence of the constraint is decreased. This loop of alternating minimization of $\hat{F}(\boldsymbol{x}, \lambda_t, \mu)$ and updating $\lambda_t$ can be repeated until a sufficiently good solution is found or the procedure converges if $\lambda_t$ does not receive updates anymore. For multiple constraints $c_j(\boldsymbol{x})$, $j = 1, \cdots, J$, the above can be readily extended with a multiplier $\lambda_t^j$ for each constraint. Since the maximization in the smooth approximation is separable in the $\lambda_t^j$, the same update rule may be applied for each $\lambda_t^j$ separately using the respective constraint $c_j(\boldsymbol{x})$.

## 4 Constrained Parameter Regularization

In this section, we introduce Constrained Parameter Regularization (CPR), where we adapt the augmented Lagrangian method to enforce upper bounds on regularization terms. Compared to classical regularization, with a fixed regularization coefficient $\gamma$, the proposed approach will allow for variable regularization coefficients $\lambda_t^j$ (Lagrange multipliers) for $j = 1, \cdots, J$ parameter matrices $\boldsymbol{\theta}^j \subseteq \boldsymbol{\theta}$ that should be regularized. These regularization coefficients are updated alongside the network parameters $\boldsymbol{\theta}$.

### 4.1 Regularization through constraints

Classical weight decay, as introduced earlier, is used as a means to restrict the freedom of parameter adaptation. This restriction is applied with a scaling factor $\gamma$ (hyperparameter) and applies uniformly to all parameters. However, we conjecture that applying an individual adaptation pressure instead may be beneficial. Unfortunately, this would require a separate coefficient for each parameter matrix where a separate weight decay should be applied. To avoid the need for separate scaling coefficients, we formulate regularization as a constrained problem. Here, the loss function $L(\boldsymbol{\theta}, \boldsymbol{X}, \boldsymbol{y})$, with network parameters $\boldsymbol{\theta}$, takes the place of the objective. Consequently, the learning problem becomes

$$\underset{\boldsymbol{\theta}}{\text{minimize}}\ L(\boldsymbol{\theta}, \boldsymbol{X}, \boldsymbol{y}) \quad \text{s.t.} \quad c_j(\boldsymbol{\theta}^j) = R(\boldsymbol{\theta}^j) - \kappa^j \leq 0, \quad \text{for} \quad j = 1, \cdots, J,$$

where $R(\boldsymbol{\theta}^j)$ is a regularization function (e.g., the squared $L_2$-norm in case of weight decay) for a parameter matrix $\boldsymbol{\theta}^j \subseteq \boldsymbol{\theta}, j = 1, \cdots, J$, and $\kappa^j \in \mathbb{R}$ denotes a chosen bound.

To solve equation 3, we follow the augmented Lagrangian method with slight modifications. First, instead of performing a full optimization of the loss before updating $\lambda_t$, we perform updates in every step. This is motivated by the fact that full optimization is generally infeasible in a deep learning setting. Moreover, similar to the difference between weight decay and $L_2$-regularization, we treat the update between the loss-dependent and the constraint-dependent part separately. Hence, instead of introducing $\hat{L}(\boldsymbol{x}, \lambda_t, \mu)$ analogously to equation 2, and performing optimization on this objective, we independently apply updates for both steps. Consequently, the constraint violations do not accumulate

**Algorithm 1** Optimization with constrained parameter regularization (CPR) .

---

**Require:** Loss Function $L(\boldsymbol{\theta}, \boldsymbol{X}, \boldsymbol{y})$ with parameters $\boldsymbol{\theta}$, and data $\mathcal{D} = \{(\boldsymbol{X}_n, \boldsymbol{y}_n)\}_{n=0}^{N}$
**Require:** Hyperparameters: Learning rate $\eta \in \mathbb{R}^+$, Lagrange multiplier update rate $\mu \in \mathbb{R}^+ (= 1.0)$
**Require:** Optimizer $\text{Opt}(\cdot)$ for minimization, Regularization function $R(\boldsymbol{\theta})$ (e.g. L2-norm)

1:   $\lambda_t^j \leftarrow 0$ for $j = 1, \cdots, J$

2:   $\kappa^j \leftarrow \text{Initialize}(\boldsymbol{\theta}_0^j)$ for $j = 1, \cdots, J$      ▷ Initializing the upper bound $\kappa$, see Section 4.3
3:   **for** $\boldsymbol{X}_t, \boldsymbol{y}_t \sim \mathcal{D}$ **do**
4:      $\boldsymbol{\theta}_{t+1} \leftarrow \boldsymbol{\theta}_t + \text{Opt}(L(\boldsymbol{\theta}_t, \boldsymbol{X}_t, \boldsymbol{y}_t), \eta)$      ▷ Classic parameter update using, e.g., Adam.
5:      **for** each regularized parameter group $\boldsymbol{\theta}_t^j$ in $\boldsymbol{\theta}_t$ **do**

6:         $\lambda_{t+1}^j \leftarrow \left( \lambda_t^j + \mu \cdot (R(\boldsymbol{\theta}_t^j) - \kappa^j) \right)^+$

7:         $\boldsymbol{\theta}_{t+1}^j \leftarrow \boldsymbol{\theta}_{t+1}^j - \nabla_{\boldsymbol{\theta}^j} R(\boldsymbol{\theta}_t^j) \cdot \lambda_{t+1}^j$

8:      **end for**
9: **end for**

---

in momentum terms. We also remove the influence of the learning rate on the regularization. From a practical perspective, our modification does not interfere with gradient-based optimization algorithms and can be readily combined with any such optimizer. The full algorithm is given by Algorithm 1.

Conceptually, the method can be understood as the $\lambda_t^j$ accumulating constraint function values (weighted with $\mu$) over the iterations $t$. These then increase (or decrease) the influence of the constraint (via its gradient) on the search direction. When points in the feasible domain are found for which $c_j(\boldsymbol{\theta}) \leq 0$, $\lambda_t^j$ decreases until it eventually reaches 0. If, on the other hand, the optimal solution lies on the boundary, where $c_j(\boldsymbol{\theta}) = 0$, $\lambda_t^j$ should converge to a value where the update direction of the optimizer and the gradient of the constraints cancel each other. However, this situation is unlikely to occur in a deep learning setting due to the stochasticity of minibatches.

### 4.2 How is CPR different from weight decay?

The optimality conditions of the CPR problem and an $L_2$-regularized training objective reveal a connection between the two approaches. To see this, consider the training objective of $L_2$ regularization with a given $\gamma$, assuming it has a minimum at $\theta^\star$. Consequently, at this point, we have $0 = \nabla L(\theta^\star) + \gamma \cdot \nabla R(\theta^\star)$, and the value of the regularization function is $R(\theta^\star)$.

If we set $\kappa^\star = R(\theta^\star)$, the Karush-Kuhn-Tucker (KKT) (optimality) conditions for CPR are $0 = \nabla L(\theta^\star) + \lambda \cdot \nabla R(\theta^\star)$ and $R(\theta^\star) - \kappa^\star \leq 0$ (which holds with equality), with the Lagrange multiplier $\lambda \geq 0$. We can see that for $\lambda^\star = \gamma$, the solution pair $(\theta^\star, \lambda^\star)$ satisfies the KKT conditions. Hence, there is a choice of $\kappa$ (namely $\kappa^\star$) for which the CPR problem has the same optimal solution candidates as the $L_2$-regularized training objective for a given $\gamma$. CPR could therefore be seen as a different approach to searching for the same solution candidates but is parameterized with different hyperparameters ($\kappa$ instead of $\gamma$). Unlike $L_2$-regularization (or weight decay), CPR can mimic the behavior of different $\gamma$ values for different parameter matrices. This behavior changes over time as the $\lambda^j$ values are updated and thus leads to different training dynamics compared to weight decay. Additionally, focusing on a bound on the regularization function $\kappa$ instead of a penalty coefficient $\gamma$ may allow us to identify better indicators for the selection of (default) values for these hyperparameters.

### 4.3 Initialization of Upper Bounds $\kappa^j$

The upper bound $\kappa$ is the most crucial hyperparameter for CPR, and we identify four ways to initialize it. (1) `Kappa-K`: Set $\kappa^j \leftarrow \kappa$ to the same value $\kappa$ for all parameter matrices. (2) `Kappa-kI`$_0$: Set $\kappa^j$ based on the initial parameter matrices' regularization function value: $\kappa^j \leftarrow k \cdot R(\boldsymbol{\theta}_{t=0}^j)$, with $k \in \mathbb{R}^+$ as the factor of the initial measure. (3) `Kappa-WS`: Train the model parameters $\boldsymbol{\theta}$ for a specific number of warm start (WS) steps $s \in \mathbb{N}^+$ and then set $\kappa^j \leftarrow R(\boldsymbol{\theta}_{t=s}^j)$. (see algorithm for

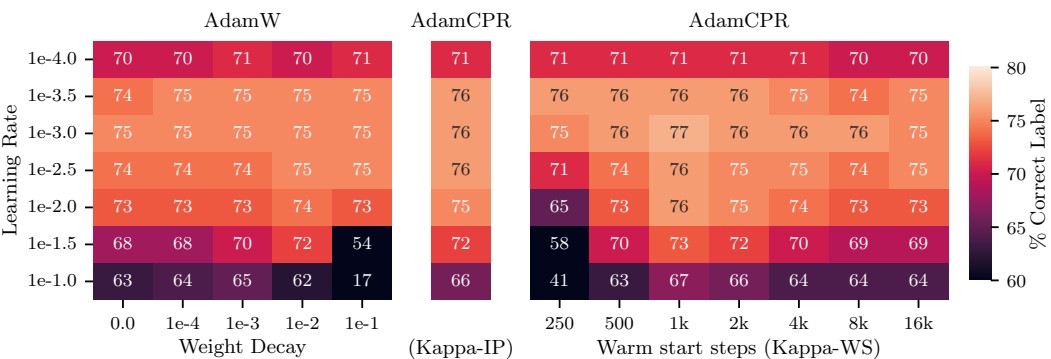

Figure 2: Percentage of correct labels (↑) of a ResNet18 trained on CIFAR100 with AdamW and AdamCPR with `Kappa-IP` or `Kappa-WS`. We use a learning rate warm-up of $500$ steps and the best `Kappa-WS` value is $2\times$ the warm-up steps. We report the mean of three runs with random seeds. We see that both CPR versions outperform weight decay

CPR with `Kappa-WS` in Appendix B). While the previous strategies all require a hyperparameter, our last strategy is essentially hyperparameter-free. (4) `Kappa-IP`: Use the first inflection point (IP) of the regularization function at step $i$ (change of curvature over the training steps) to warm start each parameter matrix individually. Specifically, $\kappa^j \leftarrow R(\boldsymbol{\theta}^j_{t=i})$ where $i$ is the first iteration where $\Delta_t \Delta_t R(\boldsymbol{\theta}^j) < 0$. The intuition behind this choice comes from the fact that the rate of change decreases at the inflection point. This hints at a saturation of the improvement expected through raising the value of the regularization function further. The position of the inflection point thus indicates a good choice for $\kappa$, as it demonstrated healthy training dynamics while still restricting the model from over-adapting (see Section 5). Consequently, this method leverages the natural progression of the model's training rather than relying on an external hyperparameter, aiming to adaptively find a suitable upper bound.

## 5 Experiments

We now describe a set of experiments to understand CPR and its parametrization. Preliminary experiments showed that $\mu$ is not a sensitive hyperparameter and we chose $\mu = 1.0$ for all our experiments. We provide a detailed analysis of $\mu$ in Appendix C. Similar to weight decay, we choose the squared $L_2$ norm as a default regularization function for CPR. We also tested an adaptive bound, where we adjusted kappa during training but found it not to be beneficial; details are reported in Appendix D. In the following experiments, we regularize all parameters in a network except for bias terms and normalization weights. Since CPR does not require additional gradient calculations or parameter updates, we find only a small runtime overhead with our CPR implementation (in PyTorch, no CUDA optimization, 0.4%-5.8% for GPT2) which is mentioned in each experiment individually and analyzed in Appendix I.

### 5.1 Train an Image Classification Model (CIFAR100)

To evaluate CPR's effectiveness and design choices, we tested AdamW and Adam with CPR (Adam-CPR) in image classification using a ResNet18 on the CIFAR100 dataset [25, 26]. We compared AdamW to AdamCPR with the four initializations described in Section 4.3. The initialization `Kappa-WS` after $s$ warm steps performed best, see Figure 2. We base our choice of the warm start on the $500$ steps learning rate warmup out of $20k$ total training steps and found a large range of hyperparameters that consistently outperform weight decay. Also, the hyperparameter-free method `Kappa-IP` outperforms weight decay. To detect the infection point, we found it sufficient to sweep the statistical measure in an interval of $10\%$ of the learning rate warmup. We also apply this in all further experiments. The superior performance of `Kappa-WS` and `Kappa-IP` may be due to its general flexibility, as warm-started bounds may be considered "learned," reflecting the actual magnitudes and distributions of the parameter matrices during training. Appendix E contains training details and a plot with all initializations and standard deviation across three random seeds in Figure E.1. ResNet18 training took 15-20 minutes on a consumer GPU, with no significant runtime difference between

Table 1: Comparison of AdamW and AdamCPR in a DeiT [28] pertaining on ImageNet. We train a small (22M parameters) and a base model (86M) with different regularization parameters.

| ImageNet Pretraining | | AdamW | | | AdamCPR | | | |
|---|---|---|---|---|---|---|---|---|
| | | weight decay | | | Kappa WS (x lr-warmup) | | | Kappa IP |
| | | 0.005 | 0.05[1] | 0.5 | 1x | 2x | 4x | |
| DeiT-Small (22M) | Top-1 Acc. (%) | 76.97 | 79.03 | 79.16 | **79.81** | 79.33 | 78.04 | **79.84** |
| DeiT-Base (86M) | Top-1 Acc. (%) | 76.19 | 78.59 | 80.56 | **81.19** | 79.61 | TBA | **80.95** |

AdamW and AdamCPR. We also tested the standard deviation as a choice for the regularization function, which performed well but not better than the squared $L_2$ norm (see Figure E.2).

To investigate the relationship between the learning rate warm-up and the number of warm start steps $s$ of `Kappa-WS` or `Kappa-IP`, we experimented with varying warm-up steps. We found that setting the CPR warm start steps $s$ to twice the warm-up steps is a good initial choice. For very low warm-up steps, the best $s$ was four times the warm-up count. Conversely, with a long warm-up phase, a shorter CPR warm start ($\times 1$) is preferable. Notably, the optimal choice of $s$ is almost independent of the learning rate, as shown in Figure E.3. The optimal warm start steps are consistent across a wide range of learning rates. A simple baseline representing a similar regularization approach is a weight decay schedule. We evaluated a cosine schedule for decreasing and increasing weight decay values, similar to [10, 11]. The results, shown in Figure E.4, indicate that the decreasing schedule outperforms a fixed weight decay but not CPR. We tested if CPR is particularly good for noisy data and perfomed experiments on the noisy CIFAR100-C dataset [27]. The results, in Figure E.5, show that AdamCPR outperforms AdamW slightly. However none of the optimizer and hyperparameter configurations lead to an outstanding performance on this task, we wouldn't claim that CPR is particularly good for noisy data. We also used CPR with SGD. We found, as shown in Figure E.6, that SGD with CPR outperforms SGD with weight decay when using the `Kappa-WS` initialization. However, `Kappa-IP` seems not to work with SGD, probably due to the changed convergence behavior in contrast to Adam.

Additionally, we compared our method to related work. We implemented AdaDecay [13] and evaluated the method for different alpha values, as seen in Figure E.7. We also compared AdamW and AdamCPR to adaptive Weight Decay (AWD) [14] and AMOS [15]. Furthermore, we used Adam with parameter rescaling from Liu et al. [18]. We found AdaDecay superior to AdamW, while AMOS and Rescaling performed less well. However, CPR outperforms all related approaches. We report all results across multiple learning rates and weight decay values in Figure E.8.

## 5.2 Train an Image Classification Model (ImageNet)

We compare AdamW and AdamCPR in vision transformer [29] training on ImageNet [30]. We choose to train the DeiT [28] model with 22M (small) and with 86M (base) parameters. We make use of the PyTorch Image Models library [31] and train with the configuration given in [28] for 300 epochs. To explore the impact of weight decay, we also train with a $10\times$ and $0.1\times$ the weight decay value. For CPR, we initialize with `Kappa-WS` ($\times$ lr-warmup) and `Kappa-IP`. We observed a minor runtime increase when using CPR. For example, training the small model on 4 A100 GPUs took 14.85h for AdamW and 14.89h for AdamCPR. All relevant hyperparameters can be found in Appendix F. As seen in Table 1, AdamCPR outperforms AdamW for small and base DeiT training with both kappa initialization methods. Most notably, the hyperparameter-free regularization with `Kappa-IP` outperforms AdamW in both cases. However, in the base model training, Kappa-WS surpasses `Kappa-IP`.

## 5.3 Fine-tuning a CLIP model

We conducted fine-tuning experiments using a CLIP model [33] on the ImageNet dataset. We used the ViT-B/32 model pre-trained by Radford et al. [33]. The model was fine-tuned for 10 epochs following the hyperparameter choices of Wortsman et al. [32] (learning rate of $3 \times 10^{-5}$, cosine-annealing learning rate schedule with 500 warm-up steps) but without the special classification head initialization and the training was performed on a single GPU with a batch size of 512. We compare

Table 2: Comparison of AdamW and AdamCPR for CLIP finetuning on ImageNet. We report the top-1 accuracy and follow the hyperparameters and schedule from WiSE-FT [32].

| ImageNet Finetuning | AdamW | | | | | AdamCPR | | | Kappa IP |
|---|---|---|---|---|---|---|---|---|---|
| | weight decay | | | | | Kappa WS | | | |
| | 0.0001 | 0.001 | 0.01 | 0.1 | 1.0 | 1x | 2x | 4x | |
| Top-1 Acc. (%) | 75.24 | 75.39 | 75.32 | 75.17 | 74.4 | 75.27 | **75.52** | 75.41 | 75.40 |

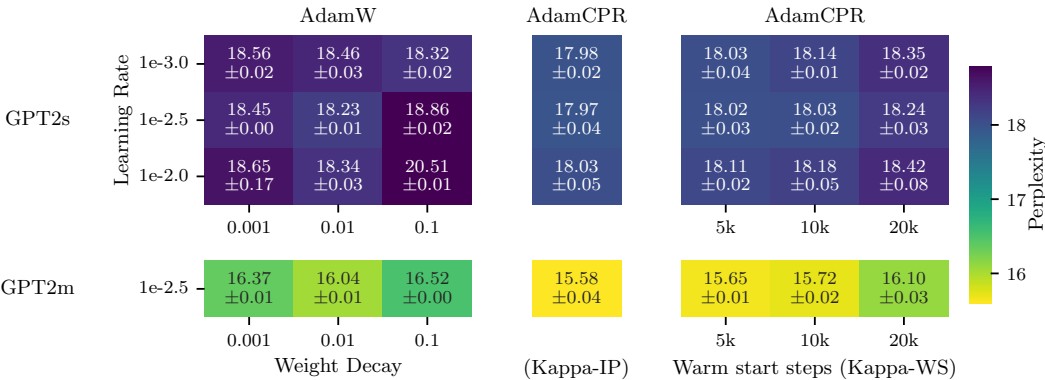

Figure 3: Perplexity ($\downarrow$) $\pm$ std across three random seeds of GPT2s and GPT2m trained on OpenWeb-Text with AdamW (left) and AdamCPR with `Kappa-IP` (middle) and AdamCPR with `Kappa-WS` (right). We use a learning rate warm-up of $5k$ steps. The CPR with the hyperparameter-free strategy `Kappa-IP` outperforms weight decay but also CPR with warm start.

AdamW with different weight decay values to AdamCPR in different configurations, where we report the top-1 accuracy after finetuning. The results in Table 2 show that the `Kappa-WS` initialization also leads to better results in this finetuning setting, comparing favorably to traditional weight decay. CPR with `Kappa-IS` performs similarly to the best weight decay values, but again, without the need for finding a regularization hyperparameter.

## 5.4 Pretraining a Large Language Model (OpenWebText)

We performed experiments training a GPT2 language model [34] on Openwebtext [35]. We compared AdamW on different weight decay values to AdamCPR using `Kappa-WS` with different warm start steps and `Kappa-IP`. We use a learning rate warmup for 5k steps ($2.5\%$ of total training steps) followed by cosine annealing. Again, we select the warm start steps of $\kappa$ based on the warmup steps of the learning rate and evaluate $s \in (5k, 10, 20k)$ steps. We train the model sizes GPT2s and GPT2m with 124M and 354M parameters for 200k steps. The results are shown in Figure 3. CPR outperforms weight decay at all learning rates, in both model sizes and with both kappa initialization strategies. We also see that the `Kappa-IP` initialized CPR runs are less sensitive to the learning rate than weight decay $\gamma$. Remarkably, CPR with the hyperparameter-free initialization `Kappa-IP` performs best, achieving $0.2$ to $0.3$ better perplexity than weight decay. To illustrate the performance difference, we trained a model with weight decay for a longer schedule to get the same performance as with CPR, the result is shown in Figure 1. CPR saves up to $33\%$ training budget on that scale. Figure 5 shows the difference in training dynamics with CPR. We find that `Kappa-IP` is close to the optimal warm start step for `Kappa-WS` but find individual starting points for different layers, see Figure G.1. We provide details of the training and hyperparameters in Appendix H. We found no runtime overhead of CPR in contrast to AdamW training GPT2s but about $2.5\%$ for GPT2m (see runtime analysis in Appendix I). We also evaluated AdaDecay [13], Adaptive Weight Decay (AWD) [14] and AMOS [15] in the GPT2s training setting but neither of the related methods outperforms AdamW nor AdamCPR, see results in Table H.1.

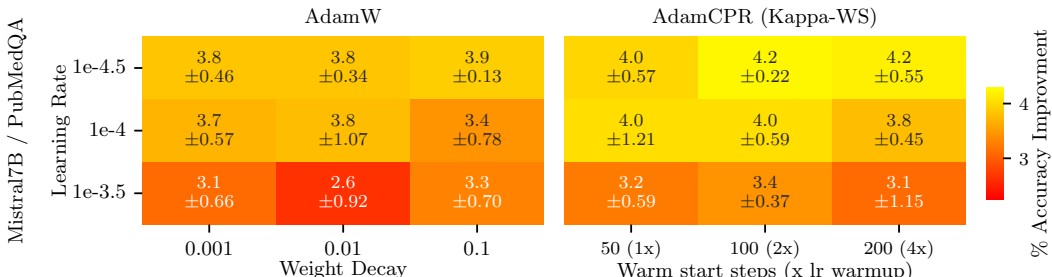

Figure 4: Percentage of performance change before and after fineuning Mistral 7B with pubmedQA artificial data (↑) with the use of AdamW (left) and AdamCPR with `Kappa-WS` (right). We use a learning rate warm-up of 50 steps. We see that CPR outperforms weight decay for each learning rate.

## 5.5 Fine-tuning a Large Language Model

Probably a more common task than pre-training a large language model (LLM) is to fine-tune one. Hence, we evaluate CPR in the fine-tuning of the Mistral7B large language model [36], incorporating low-rank adaptation (LoRA) [37]. Specifically, we fine-tune artificially generated biomedical question-answering (QA) pairs from the PubMedQA dataset [38]. We fine-tune all attention and feed-forward weights using either AdamW or AdamCPR with a learning rate warm-up of 50 steps, followed by cosine annealing. We experiment with different values of weight decay and warm start steps for `Kappa-WS`, set at $1\times$, $2\times$, and $4\times$ the learning rate warm-up steps. The fine-tuning was performed on four GPUs for about 1h. Each configuration is trained across three random seeds. We evaluate the LLM before and after the fine-tuning on the expert-annotated *PubMedQA* QA instances and report the change in answer accuracy (means and standard deviations across three random seeds) in Figure 4. The fine-tuning enhances the performance on the PubMedQA benchmark and CPR outperforms AdamW for each learning rate. As in both the ImageNet and GPT2 experiments, the best `Kappa-WS` value was $2\times$ the warm-up steps (here, $50 \times 2$). We also tested `Kappa-IP` but it performed worse due to the lack of an inflection point for some parameters, short learning rate warmup, and different training dynamics with LoRA. We also found that CPR helps to mitigate catastrophic forgetting, therefore we evaluate before and after finetuning on a set of benchmarks and found that CPR with some learning rates helps to reduce a performance drop e.g. on the *TruthfulQA* benchmark, which evaluates models' abilities to mimic human falsehoods [39], on up to $3\%$ (see results in Figure K.1). Detailed hyperparameters and plots including standard deviations are available in Appendix K.

## 5.6 Medical Image Segmentation

Aside from image classification, we also applied CPR to (medical) image segmentation using the nnU-Net framework [40] and training with the *SGD optimizer* in combination with CPR with `Kappa-WS`. For this, we considered the tasks of Multi-Atlas Labeling Beyond the Cranial Vault (BTCV) [41] where we improve the Dice score from $83.99$ to $84.23$, the Heart Segmentation task of the Medical Segmentation Decathlon [42] where we improve the Dice score from $92.92$ to $93.18$ and the 2020 version of the Brain Tumor Segmentation challenge (BraTS) task [43] where we improve the Dice score from $76.22$ to $76.65$. These results show that CPR also works in combination with SGD where we replace weight decay. Training details for the task and all results are in Appendix J.

## 6 Discussion

Our extensive evaluation of Constrained Parameter Regularization (CPR) across multiple tasks underscores its effectiveness as a robust alternative to traditional weight decay. A critical aspect of CPR's success is its initialization strategy. To this end, we propose four strategies to initialize the upper bound $\kappa$. With our findings, we identify two strategies, `Kappa-WS` and `Kappa-IP` as prime candidates showing a strong performance, consistent across multiple tasks. The good performance of the warm-started bound `Kappa-WS` can be attributed to the fact that even a carefully chosen initialization of parameters does not consider the training task and data. Therefore, the actual parameter weights during training are better reflected in a warm-started bound, which also takes into

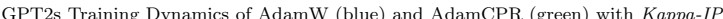

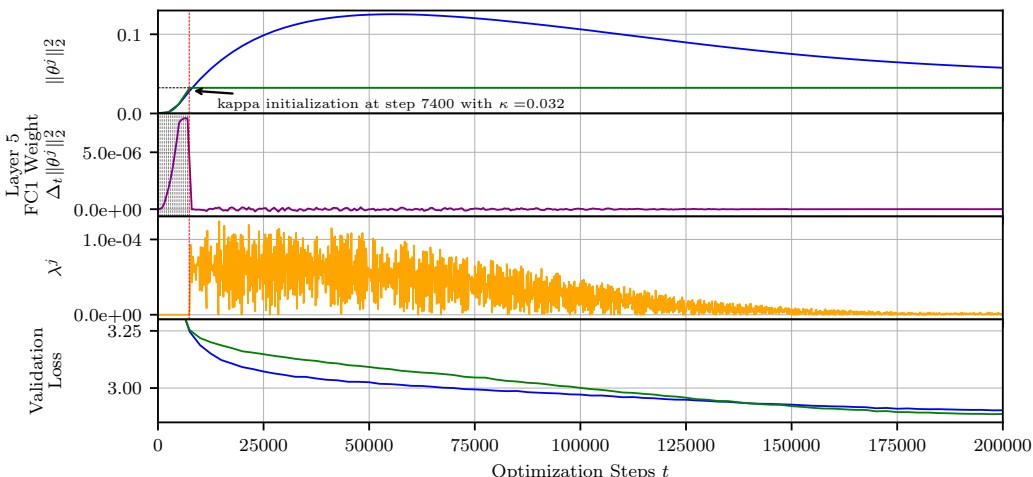

Figure 5: The training dynamics of AdamW (blue) and AdamCPR with `Kappa-IP` (green) in a GPT2s training run. The upper plot shows the squared L2 norm of the first fully connected weight in the fifth layer. Below we see the gradient of the squared L2 norm regarding the training steps. After the inflection point (7400), `Kappa-IP` initializes kappa $\kappa^j \leftarrow R(\boldsymbol{\theta}^j_{t=i})$ and starts the regularization. The third plot shows CPR's lambda enforcing the constraint. At the bottom, we see the validation loss. AdamW converges faster in the beginning of the training but CPR leads to a more linear improvement and a better final performance.

account the network's depth and the varying gradient updates in deeper layers. We found that setting the CPR warm start steps $s$ to twice the learning rate warm-up steps serves as an effective initial configuration for any training setup. However in a pre-training setting, setting the upper bound based on the first inflection point of the regularization function (`Kappa-IP`) yields an additional advantage: It removes even the one hyperparameter present in the warm start strategy, bringing the regularization capabilities of CPR without any additional hyperparameters. Simultaneously, this strategy shows best-in-class performance in GPT2 training, seemingly even extending the range of usable learning rates on a given task. This reduces the effort in hyperparameter optimization not only for the optimal regularization but also for the optimal learning rate. CPR also changes the training dynamics, as shown in Figure 5 and Figure G.1. While both weight decay and CPR can achieve a similar final L2 regularization, the path to this norm is different. Weight decay allows for intermediate overadaptation with high L2 norms, whereas CPR controls the L2 norm throughout the entire training process. This results in a slower initial loss drop but a more consistent decay, leading to a better final performance.

A noted limitation of CPR is an increase in runtime by up to 6% for larger models (1.1B parameters), as detailed in Appendix I. However, for smaller models or larger batch sizes, this overhead is negligible. The benefit of CPR diminishes in scenarios where weight regularization has minimal impact, such as when training small models on large datasets with a high ratio of training samples to parameters. Future research could explore the application of CPR to even larger models and a broader range of tasks.

## 7 Conclusion

Constrained Parameter Regularization (CPR) offers a significant advancement in regularization techniques, providing a robust and efficient alternative to traditional methods. By enforcing an upper bound on the regularization function, CPR integrates seamlessly with gradient-based optimizers and incurs minimal runtime overhead. Its dynamic tailoring of regularization to individual parameter matrices and reduces hyperparameter optimization by eliminating the need for a weight regularization hyperparameter in pre-training. Our four experiments demonstrate that neural networks trained using CPR outperform those with traditional weight decay. These findings highlight CPR's potential as a versatile and powerful tool for improving model performance and open promising future research.

## Acknowledgements

This research was funded by the Deutsche Forschungsgemeinschaft (DFG, German Research Foundation) under grant number 417962828. We acknowledge funding by the European Union (via ERC Consolidator Grant DeepLearning 2.0, grant no. 101045765). Views and opinions expressed are however those of the author(s) only and do not necessarily reflect those of the European Union or the European Research Council. Neither the European Union nor the granting authority can be held responsible for them.

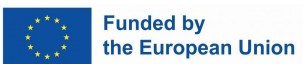

The authors gratefully acknowledge the Gauss Centre for Supercomputing e.V. (www.gauss-centre.eu) for funding this project by providing computing time on the GCS Supercomputer JUWELS [44] at Jülich Supercomputing Centre (JSC). We acknowledge the financial support of the Hector Foundation.

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

# Appendix

## A  Derivation of the Lagrange multiplier update

For simplicity, we consider a single constraint. Note that multiple constraints can be addressed separately as the optimization problem would be separable in the respective $\lambda^j$. We need to solve

$$\underset{\lambda \geq 0}{\text{maximize}} \; f(\boldsymbol{x}) + \lambda \cdot c(\boldsymbol{x}) - \frac{1}{2\mu}(\lambda - \lambda_t)^2.$$

The optimal point of this problem is equivalent to the optimal point of

$$\underset{\lambda}{\text{minimize}} \; -f(\boldsymbol{x}) - \lambda \cdot c(\boldsymbol{x}) + \frac{1}{2\mu}(\lambda - \lambda_t)^2 \quad \text{s.t.} \quad -\lambda \leq 0.$$

To find candidates for optimal points, we need to solve the Karush–Kuhn–Tucker (KKT) system with the Lagrange function $\mathcal{L}(\lambda, \psi)$ and the Lagrange multiplier $\psi$

$$\mathcal{L}(\lambda, \psi) = -f(\boldsymbol{x}) - \lambda \cdot c(\boldsymbol{x}) + \frac{1}{2\mu}(\lambda - \lambda_t)^2 - \psi \cdot \lambda$$

Which leads to the KKT system

$$\nabla_\lambda \mathcal{L}(\lambda, \psi) = 0 \iff 0 = -c(\boldsymbol{x}) + \frac{1}{\mu}(\lambda - \lambda_t) - \psi$$

$$\nabla_\psi \mathcal{L}(\lambda, \psi) \leq 0 \iff 0 \geq -\lambda$$

$$\lambda \cdot \psi = 0 \tag{3}$$

According to the complementary conditions in equation 3, the constraint is either active, hence $\lambda = 0$ and $\psi \geq 0$ or inactive, such that $\lambda > 0$, and consequently, $\psi = 0$.

**Case**: $\lambda = 0$ and $\psi \geq 0$

Here, $\lambda = 0$ (by assumption), and $\psi$ is given by

$$\nabla_\lambda \mathcal{L}(\lambda, \psi) = 0 \iff 0 = -c(\boldsymbol{x}) + \frac{1}{\mu}(0 - \lambda_t) - \psi$$

$$\psi = -c(\boldsymbol{x}) - \frac{\lambda_t}{\mu}$$

Since we require $\psi \geq 0$ for a KKT point, (note that $\mu > 0$)

$$0 \leq \psi = -c(\boldsymbol{x}) - \frac{\lambda_t}{\mu}$$

$$\iff 0 \leq -\mu \cdot c(\boldsymbol{x}) - \lambda_t$$

$$\iff 0 \geq \lambda_t + \mu \cdot c(\boldsymbol{x})$$

Consequently, $\lambda = 0$ is a candidate for the optimal point only when $0 \geq \lambda_t + \mu \cdot c(\boldsymbol{x})$.

**Case**: $\lambda > 0$ and $\psi = 0$ (inactive constraint)

For this case we get

$$\nabla_\lambda \mathcal{L}(\lambda, \psi) = 0 = -c(\boldsymbol{x}) + \frac{1}{\mu}(\lambda - \lambda_t) - 0$$

$$0 = -\mu \cdot c(\boldsymbol{x}) + \lambda - \lambda_t$$

$$\lambda = \lambda_t + \mu \cdot c(\boldsymbol{x})$$

Due to the geometry of the problem (quadratic with bound constraint), $\lambda = 0$ is the optimal solution if the constraint is active, i.e., if $\psi \geq 0$, which is the case if $0 \geq \lambda_t + \mu \cdot c(\boldsymbol{x})$. Consequently, the optimal solution is given by

$$\lambda^\star = (\lambda_t + \mu \cdot c(\boldsymbol{x}))^+. \tag{4}$$

Plugging this into $\hat{F}(\boldsymbol{x}, \lambda_t, \mu)$, we get

$$\hat{F}(\boldsymbol{x}, \lambda_t, \mu) = \begin{cases} f(\boldsymbol{x}) + c(\boldsymbol{x})(\lambda_t + \frac{\mu}{2}c(\boldsymbol{x})), & \text{if} \quad \lambda_t + \mu \cdot c(\boldsymbol{x}) \geq 0 \\ f(\boldsymbol{x}) - \frac{1}{2\mu}\lambda_t^2, & \text{else} \end{cases}$$

And the gradient with respect to $\boldsymbol{x}$ is

$$\nabla_{\boldsymbol{x}}\hat{F}(\boldsymbol{x}, \lambda_t, \mu) = \begin{cases} \nabla_{\boldsymbol{x}}f(\boldsymbol{x}) + \nabla_{\boldsymbol{x}}c(\boldsymbol{x})(\lambda_t + \mu \cdot c(\boldsymbol{x})), & \text{if} \quad \lambda_t + \mu \cdot c(\boldsymbol{x}) \geq 0 \\ \nabla_{\boldsymbol{x}}f(\boldsymbol{x}) - 0 & \text{else} \end{cases}$$

Or more compactly by using equation 4

$$\nabla_{\boldsymbol{x}}\hat{F}(\boldsymbol{x}, \lambda_t, \mu) = \nabla_{\boldsymbol{x}}f(\boldsymbol{x}) + \nabla_{\boldsymbol{x}}c(\boldsymbol{x}) \cdot \lambda^\star.$$

# B    The CPR Algorithm with `Kappa-WS`

---

**Algorithm 2** Optimization with constrained parameter regularization (CPR) and `Kappa-WS` .

---

**Require:** Loss Function $L(\boldsymbol{\theta}, \boldsymbol{X}, \boldsymbol{y})$ with parameters $\boldsymbol{\theta}$, and data $\mathcal{D} = \{(\boldsymbol{X}_n, \boldsymbol{y}_n)\}_{n=0}^N$
**Require:** Hyperparameters: Learning rate $\eta \in \mathbb{R}^+$, Lagrange multiplier update rate $\mu \in \mathbb{R}^+$, starting step $s$ for CBR.
**Require:** Optimizer $\text{Opt}(\cdot)$ for minimization, Regularization function $R(\boldsymbol{\theta})$ (e.g. L2-norm)
 1: # Initialization
 2: $t \leftarrow 0$
 3: $\boldsymbol{\theta}_t \leftarrow \text{Initialize}(L(\cdot))$
 4: $\lambda_t^j \leftarrow 0$ for $j = 1, \cdots, J$
 5: $\kappa^j \leftarrow \infty \ j = 1, \cdots, J$
 6: # Training
 7: **for** $\boldsymbol{X}_t, \boldsymbol{y}_t \sim \mathcal{D}$ **do**
 8:      $\boldsymbol{\theta}_{t+1} \leftarrow \boldsymbol{\theta}_t + \text{Opt}(L(\boldsymbol{\theta}_t, \boldsymbol{X}_t, \boldsymbol{y}_t), \eta)$          ▷ Classic parameter update using, e.g., Adam.
 9:      **for** each regularized parameter group $\boldsymbol{\theta}_t^j$ in $\boldsymbol{\theta}_t$ **do**
10:          $\lambda_{t+1}^j \leftarrow \left(\lambda_t^j + \mu \cdot (R(\boldsymbol{\theta}_t^j) - \kappa^j)\right)^+$
11:          $\boldsymbol{\theta}_{t+1}^j \leftarrow \boldsymbol{\theta}_{t+1}^j - \nabla_{\boldsymbol{\theta}^j}R(\boldsymbol{\theta}_t^j) \cdot \lambda_{t+1}^j$
12:          **if** $t = s$ **then**          ▷ `Kappa-kI`$_s$ initialization, see Section 4.3.
13:              $\kappa^j \leftarrow R(\boldsymbol{\theta}_t^j)$
14:          **end if**
15:      **end for**
16:      $t \leftarrow t + 1$
17: **end for**

---

## C  Experiments on the Sensitivity of the Update Rate $\mu$

We analyze the sensitivity of the update rate $\mu$ in CPR with experiments on ResNet18 trained on the CIFAR100 and GPT2s trained on OpenWebText. For the ResNet18 experiments, we consider update rates from $\mu = 0.01$ to $\mu = 10$ and apply two kappa initialization methods, Kappa-kI$_0$ and Kappa-WS. As shown in Figure C.1 we see no significant impact of $\mu$ on the performance. We report the mean percentage of correct labels across three random seeds. We also performed short-runtime experiments with GPT2s and update rates of $\mu \in \{0.01, 0.1, 1, 10\}$. and observe very similar results, see Table C.1. To get an impression of how $\mu$ impacts $\lambda$ and therefore the squared L2 norm in the weight matrices with the use of CPR, we plotted the squared L2 norm and $\lambda$ for three weight matrices during the training in Figure C.2. We found no impact on the stability of the squared L2 norm despite the difference in the magnitude of the $\lambda$ for different $\mu$ values.

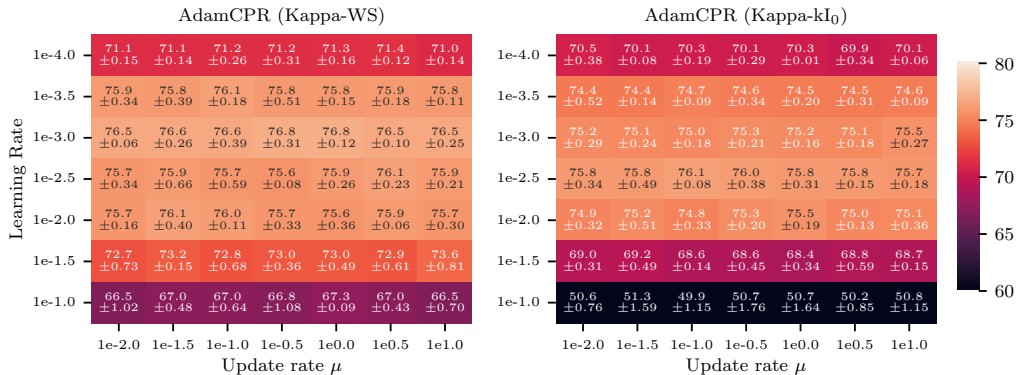

Figure C.1: The Figure shows the percentage of correct labels of the ResNet18 trained on the CIFAR100 with the use of Kappa-kI$_0$ (left), AdamCPR (Kappa-WS) (right) with different update rates $\mu$. The elements in the heat map are experiments with different learning rates and each element is colored according to the mean accuracy of three random seeds and the numbers are the mean accuracy and standard deviation of the experiments. The experiment shows that the AdamCPR regularization is not sensitive to the choice of the $\mu$ parameter.

Table C.1: Comparison of different values for the update rate $\mu$ of AdamCPR. We run experiments with GPT2s with 50k total steps, a learning rate warmup of 2.5k steps, and a kappa warm start of 5k steps.

| Method ($\mu$ value) | Accuracy ↑ | PPL ↓ |
|---|---|---|
| **GPT2s** | | |
| AdamCPR $\mu = 10$ | 0.422 | 20.91 |
| AdamCPR $\mu = 1$ | 0.423 | 20.90 |
| AdamCPR $\mu = 0.1$ | 0.423 | 20.90 |
| AdamCPR $\mu = 0.01$ | 0.423 | 20.90 |

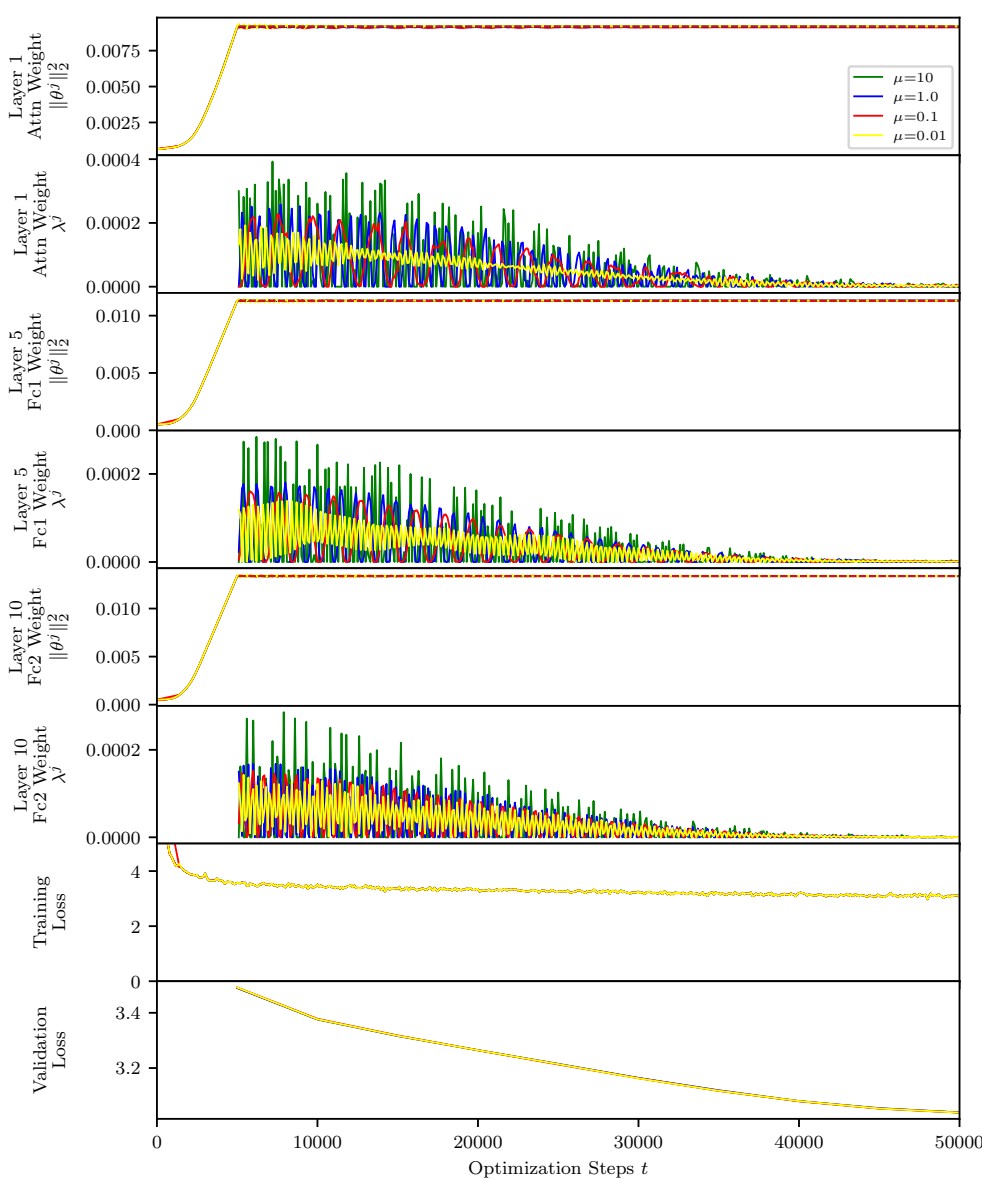

Figure C.2: A comparison of different $\lambda$ update rates $\mu$ in the training of a GPT2s model. We see three weight matrices during the training with AdamCPR. We also see how $\lambda$ regulates the constraint of the bound on the squared L2 norm. The bottom two plots show the training and validation loss.

# D  Adaptive Bounds

With fixed bounds $\kappa^j$, some parameter matrices $\boldsymbol{\theta}^j$, for which $\lambda_t^j = 0$ will not be regularized. While this can be beneficial, CPR can also be used to apply continuous pressure similar to weight decay. For this, the bounds $\kappa^j$ of parameter matrices $\boldsymbol{\theta}^j$ with $\lambda^j = 0$ can be set to the current value of the constraint function $\kappa_{t+1}^j \leftarrow c(\boldsymbol{\theta}_t^j)$. Such an adaption guarantees that each parameter matrix is always exposed to some regularization. This should result in a gradual reduction of the bounds $\kappa^j$ throughout training without exerting excessive pressure on the optimization process. In our experiments, we refer to the usage of adaptive bounds as *AdaCPR*.

This contrasts with weight decay, where continuous pressure is applied to enhance generalization throughout the training. To emulate the continuous pressure of weight decay, we propose an adaptive mechanism to adjust the upper regularization bound during training. This can be achieved by leveraging existing states. Specifically, the value of $\lambda^j$ offers insights into constraint violations. When $\lambda^j = 0$, the constraint $c_j(\boldsymbol{\theta})$ can be regarded as inactive. In this case, we may consider adjusting its bound $\kappa^j$ to align with the current constraint value of $c(\boldsymbol{\theta}_j)$. To implement these adaptive bounds, we add a conditional update rule for $\kappa^j$ after our CPR update. It updates the upper bound for each parameter matrix $\theta_t^j$ individually by

$$\kappa_{t+1}^j \leftarrow \begin{cases} R(\theta_t^j) & \text{if } \lambda_t^j = 0 \text{ and } \lambda_{t-1}^j > 0 \\ \kappa_t^j & \text{otherwise,} \end{cases}$$

where $\lambda_{t-1}^j > 0$ indicates that the upper bound was previously violated and $c_j(\boldsymbol{\theta}^j)$ was active. Consequently, this enables a gradual reduction of the bounds $\kappa^j$ throughout training without exerting excessive pressure on the optimization process. Please find AdaCPR in Algorithm 3 below.

---

**Algorithm 3** Optimization with adaptive bound constrained parameter regularization ( Ada CPR ).

---

**Require:** Loss Function $L(\boldsymbol{\theta}, \boldsymbol{X}, \boldsymbol{y})$ with parameters $\boldsymbol{\theta}$, and data $\mathcal{D} = \{(\boldsymbol{X}_n, \boldsymbol{y}_n)\}_{n=0}^N$
**Require:** Hyperparameters: Learning rate $\eta \in \mathbb{R}^+$, Lagrange multiplier update rate $\mu \in \mathbb{R}^+$
**Require:** Optimizer $\text{Opt}(\cdot)$ for minimization, Regularization function $R(\boldsymbol{\theta})$ (e.g. L2-norm)
  1: # Initialization
  2: $t \leftarrow 0$
  3: $\boldsymbol{\theta}_t \leftarrow \text{Initialize}(L(\cdot))$
  4: $\lambda_t^j \leftarrow 0$ for $j = 1, \cdots, J$
  5: $\kappa^j \leftarrow \boldsymbol{\theta}_t^j - \text{Initialize}(\boldsymbol{\theta}_0^j)$ for $j = 1, \cdots, J$
  6: # Training
  7: **for** $\boldsymbol{X}_t, \boldsymbol{y}_t \sim \mathcal{D}$ **do**
  8:     $\boldsymbol{\theta}_{t+1} \leftarrow \boldsymbol{\theta}_t + \text{Opt}(L(\boldsymbol{\theta}_t, \boldsymbol{X}_t, \boldsymbol{y}_t), \eta)$     ▷ Classic parameter update using, e.g., Adam.
  9:     **for** each regularized parameter group $\boldsymbol{\theta}_t^j$ in $\boldsymbol{\theta}_t$ **do**
 10:         $\lambda_{t+1}^j \leftarrow \left(\lambda_t^j + \mu \cdot (R(\boldsymbol{\theta}_t^j) - \kappa^j)\right)^+$
 11:         $\boldsymbol{\theta}_{t+1}^j \leftarrow \boldsymbol{\theta}_{t+1}^j - \nabla_{\boldsymbol{\theta}^j} R(\boldsymbol{\theta}_t^j) \cdot \lambda_{t+1}^j$
 12:         **if** $\lambda_t^j = 0$ and $\lambda_{t-1}^j > 0$ **then**     ▷ Update $\kappa^j$ if the constraints are not active.
 13:             $\kappa^j \leftarrow R(\boldsymbol{\theta}_t^j)$
 14:         **end if**
 15:     **end for**
 16:     $t \leftarrow t + 1$
 17: **end for**

---

The experimental results in Figure E.1 also show that the adaptation of the upper bound during the training is not beneficial. While it does not harm the performance, it also does not lead to a substantial improvement. We therefore do not use it to keep our method as simple as possible.

# E   Experiments on Image Classification (CIFAR100)

For the $\kappa$ initialization `Kappa-K`, we use a range of $\kappa = [0.005, \ldots, 0.16]$, for `Kappa-kI`$_0$ a range of $k = [4, \ldots, 256]$, and for `Kappa-WS` a range of $s = [250, \ldots, 4000]$ steps. We use a learning rate warmup of $500$ steps followed by a closing annealing. This is $2.5\%$ of the total training steps (20k). For a detailed list of training hyperparameters, we refer the reader to Table E.1.

We found that initializing with `Kappa-kI`$_0$ performs better than selecting a uniform $\kappa$ in `Kappa-K`. This may be explained by the value of the regularization function depending on the size of the jointly regularized parameter matrix and initialization method. The warm start $\kappa$ initialization method, `Kappa-WS`, performed the best. The best configuration with CPR outperforms weight decay and the choice of hyperparameters seems to be more robust.

Table E.1: Hyperparameters of the ResNet18 on CIFAR100 experiment.

| Parameter | Value |
| --- | --- |
| Seed | 1,2,3 |
| Dataset | CIFAR100 |
| Batch size | 128 |
| Training Steps | 20000 |
| Model | ResNet18 |
| Optimizer | AdamW / Adam+Rescaling / AdamCPR |
| Learning Rate | 0.001 |
| Beta1 | 0.9 |
| Beta2 | 0.98 |
| Weight Decay | 0.1 |
| Lr Schedule | Cosine with warmup |
| Lr Warmup Steps | 500 |
| Lr Decay Factor | 0.1 |
| Rescale Alpha | $0, 0.8 \ldots 16$ |
| CPR$-\mu$ | 1.0 |
| CPR-$\kappa$ | $0.8 \ldots 16$ |
| CPR-$k$ | $4 \ldots 256$ |
| CPR-$\kappa$ warm-start steps | $250 \ldots 16000$ |
| Adaptive Bounds | False / True |

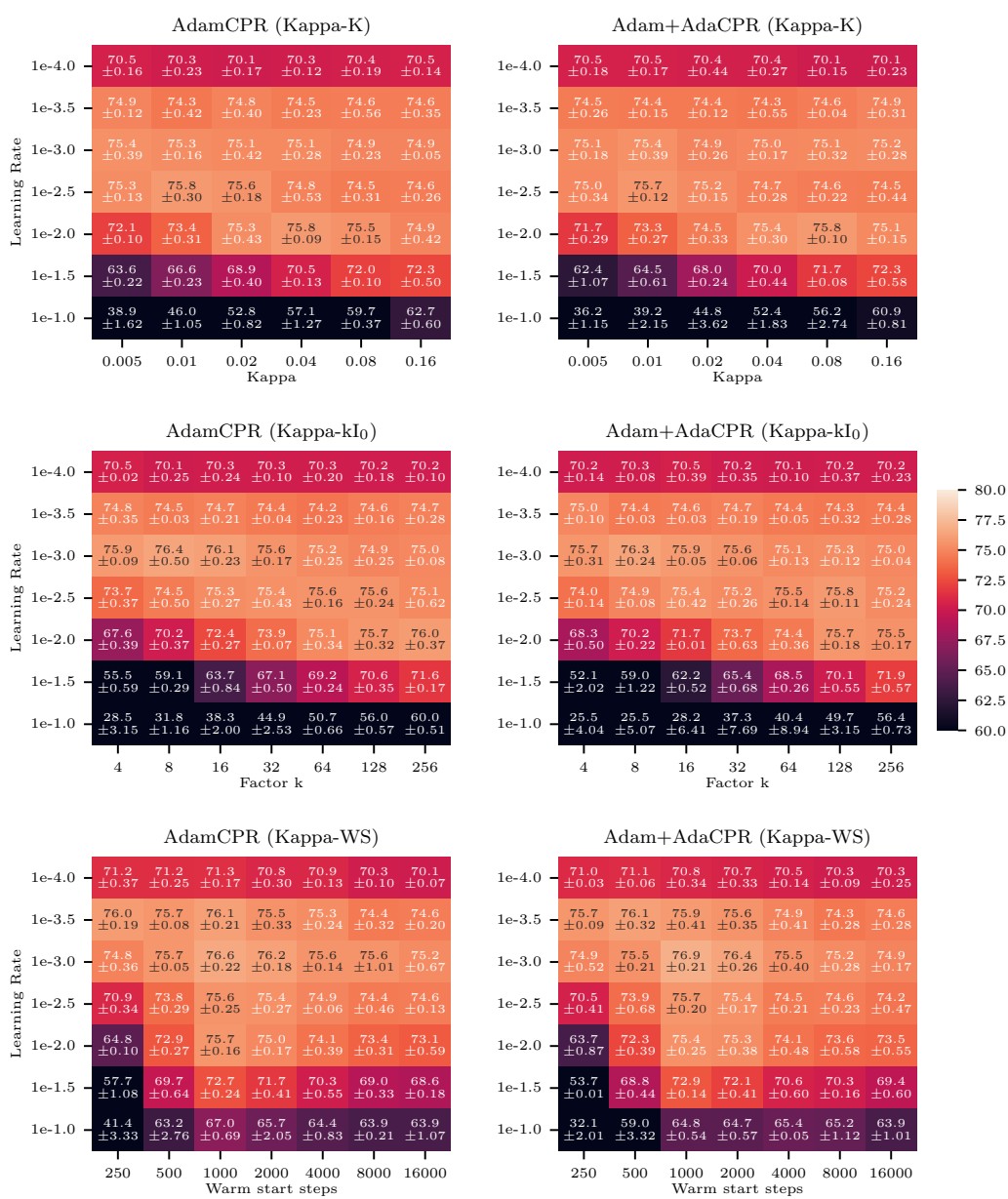

Figure E.1: Percentage of correct labels of the ResNet18 trained on the CIFAR100 with use of Adam with CPR (left) and AdaCPR (right) with use of the three different initialization techniques from Section 4.3, from top to bottom: Kappa-K, Kappa-kI$_0$, and Kappa-WS. The elements in the heat map are experiments with different learning rates and regularization hyperparameters. Each element is colored according to the mean accuracy of three random seeds and the numbers are the mean accuracy and standard deviation of the experiments.

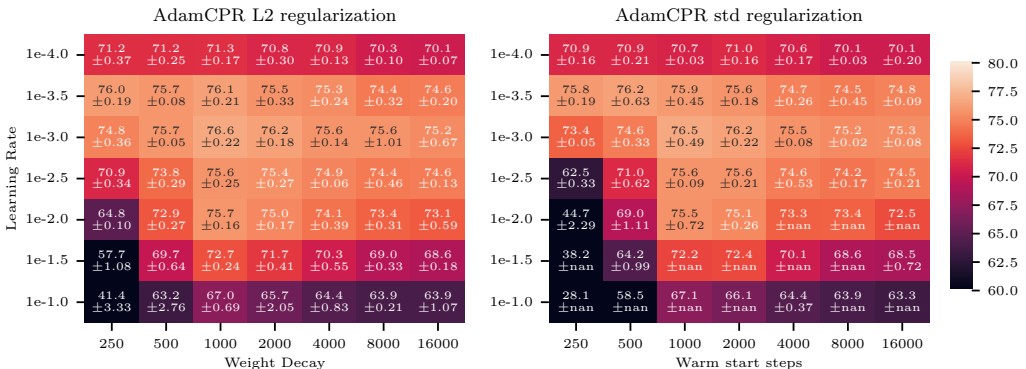

Figure E.2: Percentage of correct labels of the ResNet18 trained on the CIFAR100 with the use of AdamCPR using L2 regularization measure (left) and standard deviation as regularization measure (right). The elements in the heat map are experiments with different learning rates and warm start steps ($s$ of Kappa-WS). Each element is colored according to the mean accuracy of three random seeds and the numbers are the mean accuracy and standard deviation of the experiments.

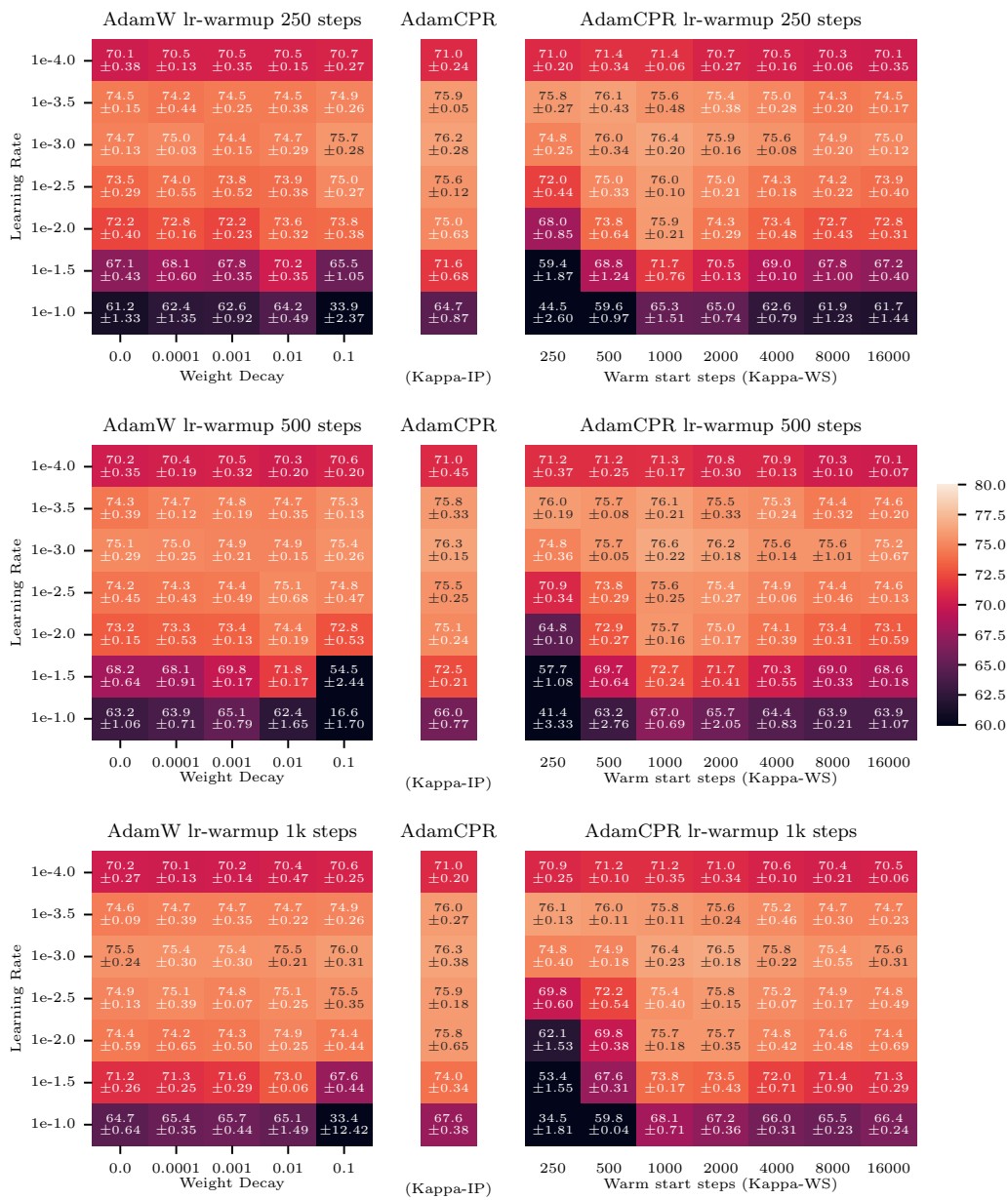

Figure E.3: Comparison of AdamW and AdamCPR with different learning rate warm-up steps. The Figure shows the percentage of correct labels of the ResNet18 trained on the CIFAR100 with the use of AdamW (left side), AdamCPR (Kappa-IP) (middle), and AdamCPR (Kappa-WS) (right side) with learning rate warm-up steps between 250 and 1000 steps. The elements in the heat map are experiments with different learning rates and regularization hyperparameters. Each element is colored according to the mean accuracy of three random seeds and the numbers are the mean accuracy and standard deviation of the experiments.

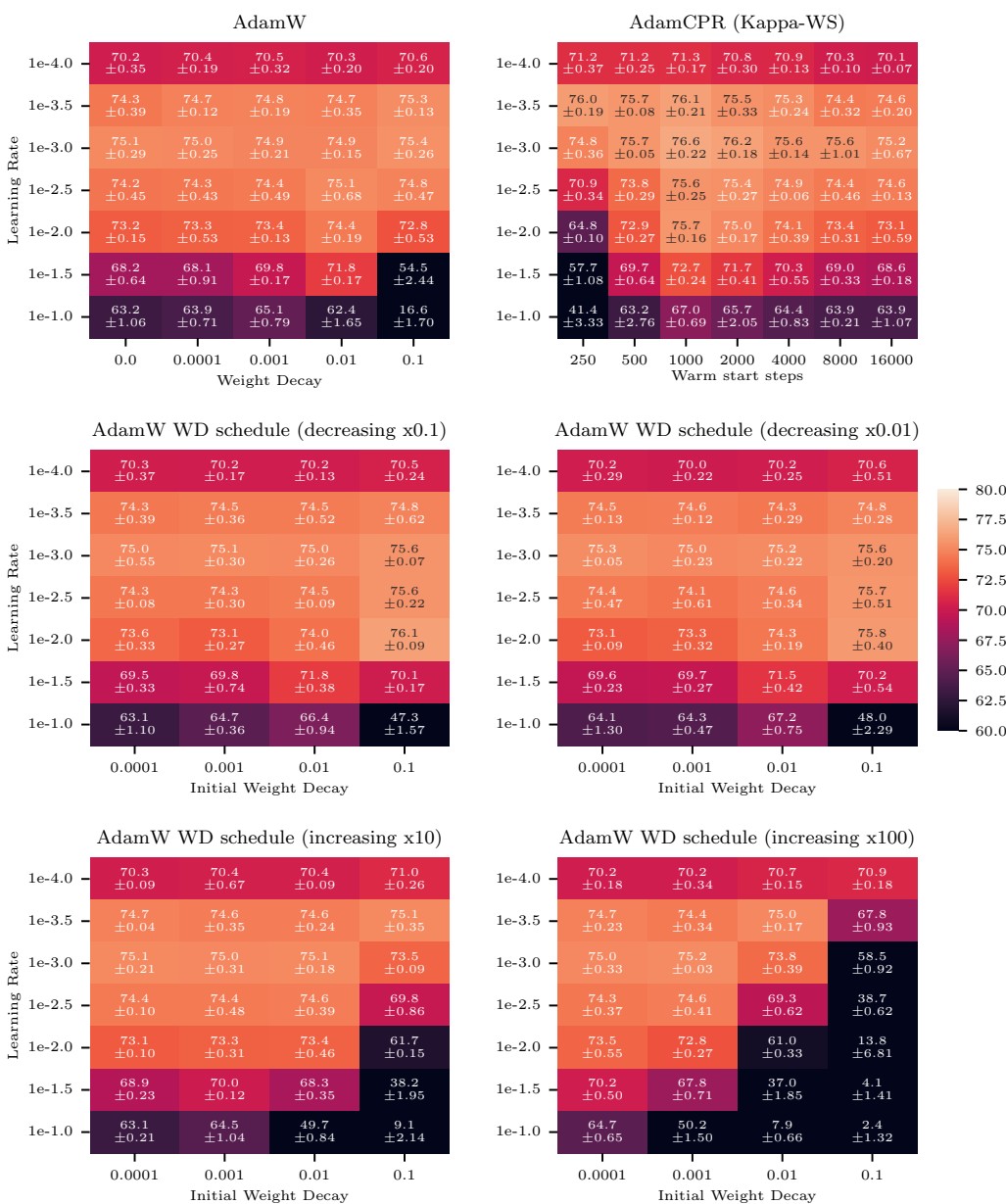

Figure E.4: Comparison of AdamW, AdamCPR, and weight decay scheduling similar to [10, 11]. The Figure shows the percentage of correct labels of the ResNet18 trained on the CIFAR100 with the use of AdamW (top left), AdamCPR (`Kappa-WS`) (top right), and Adam with weight decay scheduling. We evaluated the task with cosine decreasing weight decay to 0.1 and 0.01 times the initial weight decay value and with cosine increasing weight decay to 10 and 100 times the initial weight decay value. The elements in the heat map are experiments with different learning rates and regularization hyperparameters. Each element is colored according to the mean accuracy of three random seeds and the numbers are the mean accuracy and standard deviation of the experiments. It should be mentioned that Yun et al. [9] also performed weight decay scheduling on CIFAR100 with the use of a ResNet18. Since their code was not published, we point to Figure 3 of their experimental results, where an accuracy of around 60% was reported, which is below our AdamW baseline.

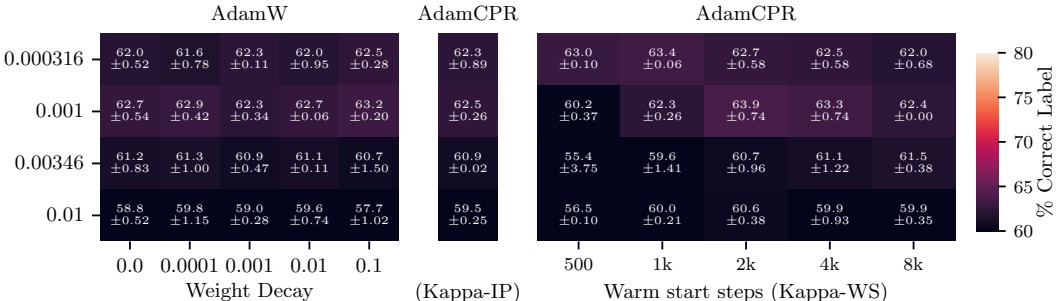

Figure E.5: Percentage of correct labels of the ResNet18 trained on the CIFAR100-C with use of AdamW (left), AdamCPR with `Kappa-IP` (middle) and AdamCPR with `Kappa-WS` (right). The elements in the heat map are experiments with different learning rates and regularization hyperparameters. Each element is colored according to the mean accuracy of three random seeds and the numbers are the mean accuracy and standard deviation of the experiments. We see that AdamCPR outperforms AdamW which could indicate that CPR leads to a more robust optimization. We see that AdamCPR performs better than AdamW with Kappa-WS but not with Kappa-IP. Kappa-IP does not fail and performs better than the average weight decay performance. None of the optimizer and hyperparameter configurations lead to an outstanding performance on this task, we wouldn't claim that CPR is particularly good for noisy data.

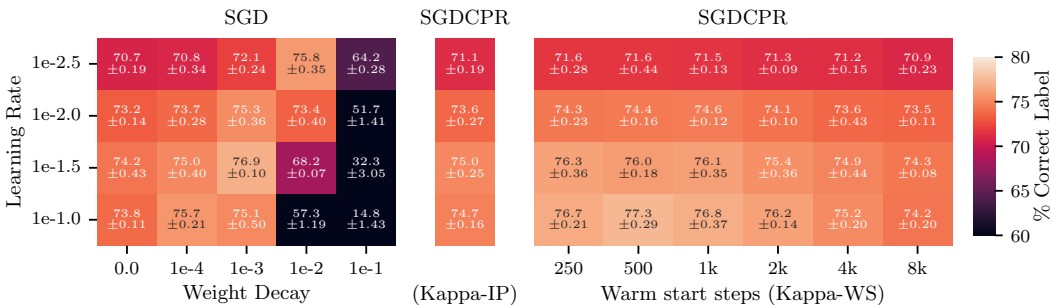

Figure E.6: Percentage of correct labels of the ResNet18 trained on the CIFAR100 with use of SGD with weight decay (left), SGD with CPR and `Kappa-IP` (middle) and SGD with CPR and `Kappa-WS` (right). The elements in the heat map are experiments with different learning rates and regularization hyperparameters. Each element is colored according to the mean accuracy of three random seeds and the numbers are the mean accuracy and standard deviation of the experiments.

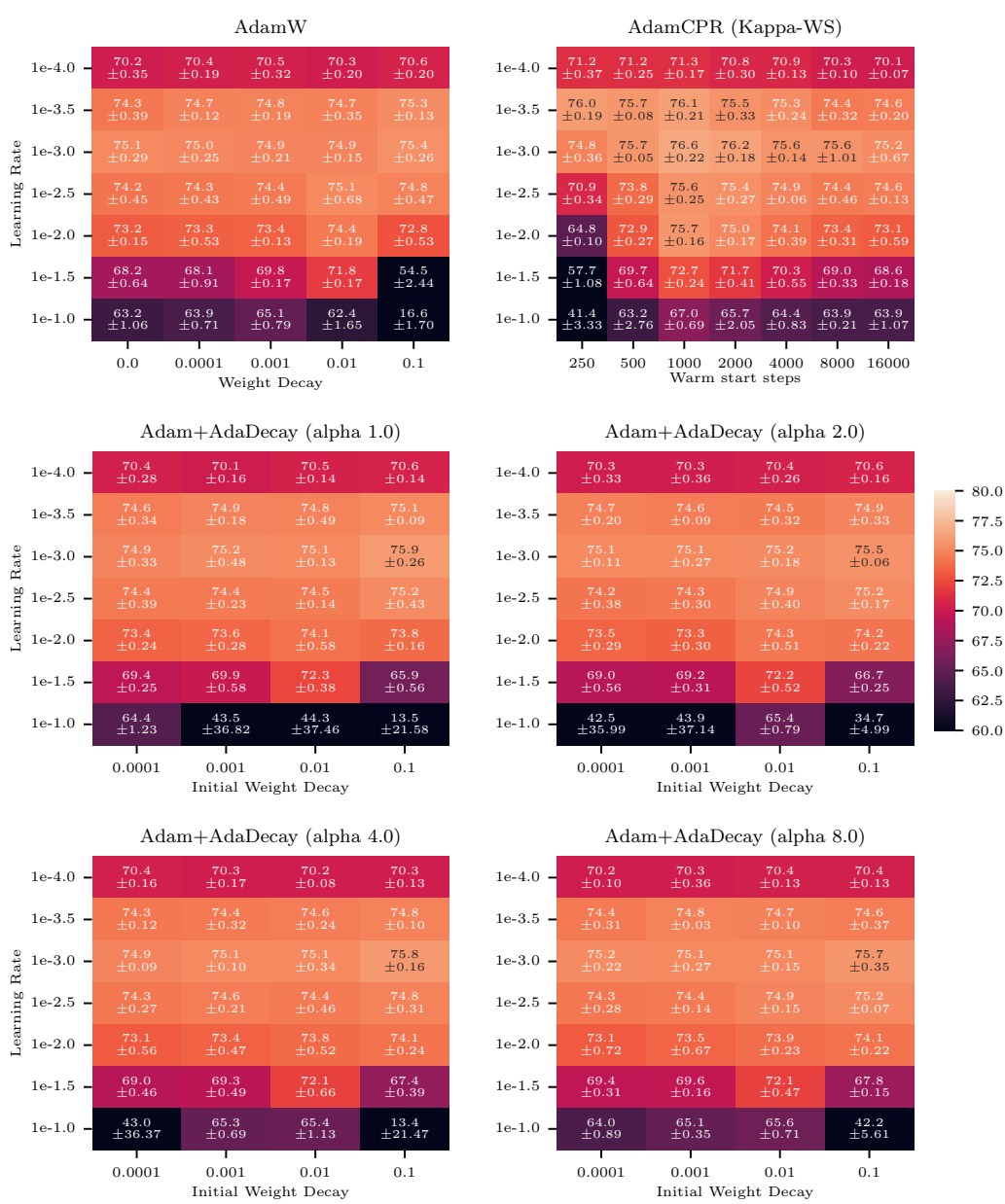

Figure E.7: Comparison of AdamW, AdamCPR, and Adam with AdaDecay [13]. The Figure shows the percentage of correct labels of the ResNet18 trained on the CIFAR100 with the use of AdamW (top left), AdamCPR (Kappa-WS) (top right), and Adam with AdaDecay with different (1.0, 2.0, 4.0, 8.0) values for the alpha hyperparameter in AdaDecay. The elements in the heat map are experiments with different learning rates and regularization hyperparameters. Each element is colored according to the mean accuracy of three random seeds and the numbers are the mean accuracy and standard deviation of the experiments.

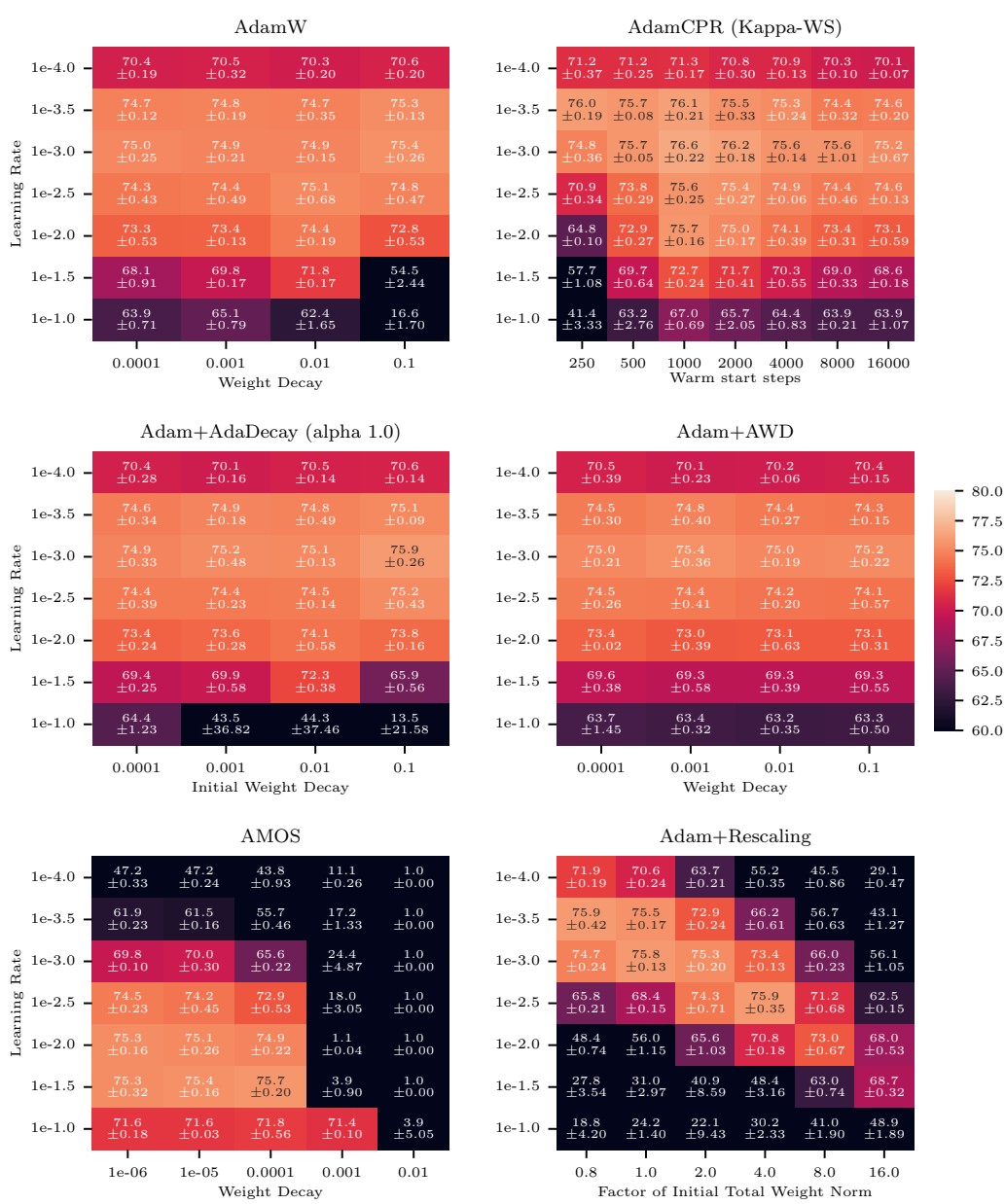

Figure E.8: Percentage of correct labels of the ResNet18 trained on the CIFAR100 with AdamW, AdamCPR, AdaDecay [13], AWD [14], AMOS [15], and Rescaling. We use different values of weight decay for AdamW, AdaDecay, AWD, and AMOS. For Adam with Rescaling, we use different factors of the initial total weight norm. AdamCPR uses `Kappa-WS`. We use a learning rate warm-up of 500 steps and the best `Kappa-WS` value is $2\times$ the warm-up steps. Each element is colored according to the mean accuracy of three random seeds and the numbers are the mean accuracy and standard deviation of the experiments.

# F   Experiments on Image Classification (ImageNet)

Table F.1: Hyperparameters for the DeiT small experiments on ImageNet.

| **ImageNet Pretraining** | AdamW | | | AdamCPR | | | |
|---|---|---|---|---|---|---|---|
| | weight decay | | | Kappa WS (x lr-warmup) | | | Kappa IP |
| | 0.005 | 0.05 | 0.5 | 1x | 2x | 4x | |
| Model Architecture | DeiT-Small (patch size 16, image size 224) | | | | | | |
| Learning Rate | 1e-3 | | | | | | |
| Warmup Epochs | 5 | | | | | | |
| Epochs | 300 | | | | | | |
| Batch Size | 256 | | | | | | |
| Optimizer | AdamW | | | AdamCPR | | | |
| Weight Decay | 0.005 | 0.05 | 0.5 | - | | | |
| $\kappa$ Init Param | - | | | 6280 | 12560 | 25120 | - |
| $\kappa$ Init Method | - | | | warm_start | | | - |
| Scheduler | cosine | | | | | | |
| Auto-augment | rand-m9-mstd0.5 | | | | | | |
| Mixup Alpha | 0.8 | | | | | | |
| CutMix Alpha | 1.0 | | | | | | |
| Random Erase Prob | 0.25 | | | | | | |
| AMP | Yes | | | | | | |
| TorchScript | Yes | | | | | | |
| Pin Memory | Yes | | | | | | |
| Data Parallel Jobs | 8 | | | | | | |

Table F.2: Hyperparameters for the DeiT base experiments on ImageNet.

| **ImageNet Pretraining** | AdamW | | | AdamCPR | | | |
|---|---|---|---|---|---|---|---|
| | weight decay | | | Kappa WS (x lr-warmup) | | | Kappa IP |
| | 0.005 | 0.05 | 0.5 | 1x | 2x | 4x | |
| Model Architecture | DeiT-Base (patch size 16, image size 224) | | | | | | |
| Learning Rate | 1e-3 | | | | | | |
| Warmup LR | 1e-6 | | | | | | |
| Min LR | 1e-5 | | | | | | |
| Warmup Epochs | 5 | | | | | | |
| Epochs | 300 | | | | | | |
| Batch Size | 256 | | | | | | |
| Optimizer | AdamW | | | AdamCPR | | | |
| Weight Decay | 0.005 | 0.05 | 0.5 | - | | | |
| $\kappa$ Init Param | - | | | 6280 | 12560 | 25120 | - |
| Drop Path Rate | 0.1 | | | | | | |
| Mixup Alpha | 0.8 | | | | | | |
| CutMix Alpha | 1.0 | | | | | | |
| Color Jitter Factor | 0.3 | | | | | | |
| Random Erase Prob | 0.25 | | | | | | |
| Train Interpolation | Bicubic | | | | | | |

# G Training Dynamics of GPT2

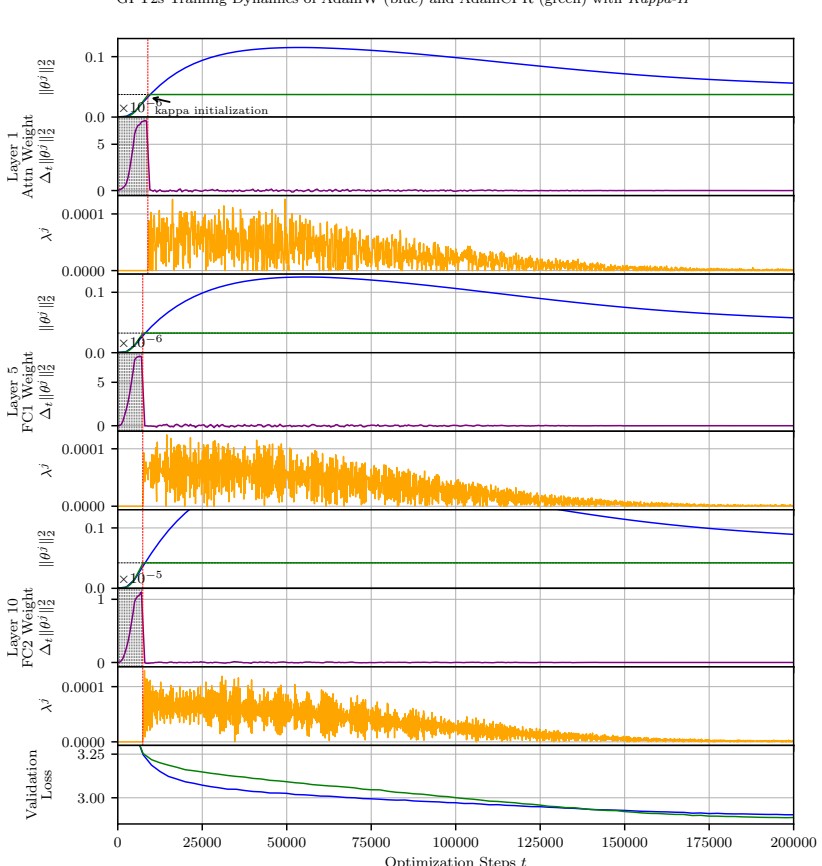

Figure G.1: The training dynamics of AdamW and AdamCPR with `Kappa-IP` of one layer in a GPT2s training run. The upper plot shows the squared L2 norm of the attention weight in the first layer. Below we see the gradient of the squared L2 norm regarding the training steps, after the first inflection point `Kappa-IP` initializes kappa and starts the regularization. The third plot shows CPR's lambda enforcing the constraint on kappa. The six plots below show the dynamics for the first weight matrix of the feed-forward block in the 5th layer and the second weight matrix of the feed-forward block in the 10th layer. At the bottom, we see the validation loss. We see that `Kappa-IP` initializes different layers at different time steps, e.g. layer 5 FC1 before layer 1 attention weights. While weight decay leads to a steady increase of the squared L2 norm for the first quarter of the training, CPR regularizes much earlier and avoids over-adaption. AdamW converges faster in the beginning of the training but CPR leads to a more linear improvement and a better final performance.

# H  Experiments on Language Modelling

For an efficient implementation, we use flash attention [45] and rotary position embedding [46]. The complete hyperparameters can be found in Appendix H. The GPT2s and GPT2m models are trained on 8 A100 GPUs up to 28h. A detailed runtime analysis can be found in Appendix I

Table H.1: Comparison of AdamW, AdamCPR, AdaDecay, AWD, and AMOS on GPT2s trained on OpenWebText. For AdamW and AdamCPR we report the mean across three random seeds. For the other methods, only a single seed is reported. The number next to the optimizer name is the weight decay coefficient $\gamma$ except for AdamCPR, here it is the number of warm start steps $s$ for `Kappa-WS`.

| Method | | Perplexity ↓ |
|---|---|---|
| AdamW | 1e-3 | $18.45 \pm 0.0039$ |
| | 1e-2 | $18.23 \pm 0.0113$ |
| | 1e-1 | $18.86 \pm 0.0169$ |
| AdamCPR (Kappa-WS) | 5k  (1x) | $18.02 \pm 0.0258$ |
| | 10k (2x) | $18.03 \pm 0.0178$ |
| | 20k (4x) | $18.24 \pm 0.0320$ |
| **AdamCPR (Kappa-IP)** | | **17.94** |
| Adam Adadecay | 1e-3 | 18.42 |
| | 1e-2 | 18.24 |
| | 1e-1 | 18.87 |
| Adam AWD | 1e-3 | 18.42 |
| | 1e-2 | 18.47 |
| | 1e-1 | 18.49 |
| AMOS | 1e-3 | NaN |
| | 1e-2 | NaN |
| | 1e-1 | NaN |

Table H.2: Hyperparameters of the language modeling task (GPT2 and Openwebtext).

| Parameter | GPT2s | GPT2m |
|---|---|---|
| GPUs | 8x A100 40GB | |
| Gradient Clip Val | 1.0 | |
| Max Steps | 200k | |
| Precision | bf16-mixed | |
| Seed | 1234 | |
| Beta1 | 0.9 | |
| Beta2 | 0.99 | |
| Eps | $1.0 \times 10^{-9}$ | |
| Bias Weight Decay | False | |
| Normalization Weight Decay | False | |
| Lr Num Warmup Steps | 5000 | |
| Lr Decay Factor | 0.1 | |
| Lr Schedule | Cosine | |
| Model Dimension | 768 | 1024 |
| Number of Layers | 12 | 24 |
| Number of Heads | 12 | 16 |
| Fed Forward Dim | 3072 | 4048 |
| Attn Dropout | 0.1 | |
| Resi Dropout | 0.1 | |
| Embed Dropout | 0.1 | |
| Rotary Pos Embed | True | |
| Rotary Emb Fraction | 0.5 | |
| Softmax Scale | True | |
| Use Bias | True | |
| Flash Attn | True | |
| Initializer | 0.02 Uniform | |
| Dataset Name | Openwebtext | |
| Max Sample Len | 1024 | |
| Batch Size | 32 | 24 |
| Val Ratio | 0.0005 | |

# I  Runtime Analysis on LLM training

To analyze the runtime in more detail, we measured the runtime per step of different regularization techniques on different GPT2 model sizes (see Table I.1). For AdamW we use the PyTorch 2.1 default implementation, for AdamCPR we adapt the AdmW implementation of PyTorch with the implementation described in Algorithm 1, for AWD and AdaDecay exists no open source implementation and we implemented it based on the PyTorch Adam class but without "for_each" optimization, and for AMOS we used the implementation form the *pytroch-optimizer* package [47]. We compare the runtime on a node with 4 A100 GPUs and report the mean time per training step across two random seeds and 3000 steps per experiment. In Table I.2 we compare the runtime with a batch size of 1 and in Table I.3 we repost the runtime with the maximal possible batch size on a 40GB A100 (in samples steps of 4).

Table I.1: GPT-2 Model Sizes and Parameter Counts

| Model | Parameters | Model Dimension | Layers | Heads |
|-------|-----------|-----------------|--------|-------|
| GPT2s | 124M | 768 | 12 | 12 |
| GPT2m | 354M | 1024 | 24 | 16 |
| GPT2l | 773M | 1280 | 36 | 20 |
| GPT2xl | 1.19B | 1600 | 36 | 25 |

Table I.2: Comparison of optimizer and regularizer runtime per step (batch size=1) across different GPT2 model sizes. Percentages indicate the increase in runtime compared to AdamW. The time is calculated as the mean time per training step across two random seeds and 3000 steps per experiment.

| Method | GPT2s | GPT2m | GPT2l | GPT2xl |
|--------|-------|-------|-------|--------|
| AdamW | 0.069s | 0.152s | 0.273s | 0.341s |
| AdamCPR | 0.073s (+5.76%) | 0.162s (+6.45%) | 0.289s (+6.09%) | 0.36s (+5.83%) |
| Adam AdaDecay | 0.111s (+60.94%) | 0.231s (+51.72%) | 0.421s (+54.51%) | 0.531s (+55.91%) |
| Adam AWD | 0.089s (+30.04%) | 0.18s (+18.55%) | 0.318s (+16.64%) | 0.385s (+13.05%) |
| AMOS | 0.146s (+113.25%) | 0.295s (+93.95%) | 0.471s (+72.61%) | 0.537s (+57.68%) |

Table I.3: Comparison of optimizer runtime per step at maximum batch size across different GPT2 model sizes. Percentages indicate the increase in runtime compared to AdamW. The time is calculated as the mean time per training step across two random seeds and 3000 steps per experiment.

| Method | GPT2s | GPT2m | GPT2l | GPT2xl |
|--------|-------|-------|-------|--------|
| AdamW | 0.25s | 0.493s | 0.473s | 0.382s |
| AdamCPR | 0.249s (-0.40%) | 0.505s (+2.44%) | 0.49s (+3.59%) | 0.404s (+5.76%) |
| Adam AdaDecay | 0.309s (+23.60%) | 0.577s (+17.05%) | 0.617s (+30.44%) | 0.573s (+50.00%) |
| Adam AWD | 0.269s (+7.60%) | 0.528s (+7.10%) | 0.517s (+9.30%) | 0.431s (+12.83%) |
| AMOS | 0.302s (+20.80%) | 0.614s (+24.54%) | 0.703s (+48.62%) | 0.581s (+52.09%) |

The runtime comparison across various GPT2 models shows that AdamCPR closely matches AdamW's efficiency, particularly at larger batch sizes where its runtime increase becomes minimal or even slightly better. In contrast, Adam AdaDecay, AWD, and AMOS significantly increase runtime, particularly in larger models and batch sizes.

However, since not all operations for CPR are implemented in a "for_each" optimized manner, CPR's runtime could benefit from an additional CUDA-optimized implementation.

# J    Experiments on Medical Image Segmentation

To demonstrate the effectiveness of the proposed CPR approach where using SGD, we also evaluate it in the context of medical image segmentation. We test CPR on four segmentation benchmarks. First, with the Adam optimizer on the Multi-Atlas Labeling Beyond the Cranial Vault (BTCV) [41] task, the Heart Segmentation task of the Medical Segmentation Decathlon [42] and the 2020 version of the Brain Tumor Segmentation challenge (BraTS) task [43].

Here, we make use of the data pipeline and network architectures following the nnU-Net framework [40], which is regarded as the state-of-the-art framework for medical image segmentation. We implement a training schedule with a total of 25k steps (for the Heart and BraTS tasks) and 125k steps for BTCV. We introduce a learning rate warmup of 2k steps ($8\%$), followed by a polynomial annealing, see all hyperparameters in Appendix J. We run each experiment on one consumer GPU for up to 2 days. We present the results in Table J.1, where different weight decay configurations in AdamW are evaluated to AdamCPR with `Kappa-WS` initialization. We report the commonly used Dice scores, averaged across cross-validation folds. These results indicate that CPR surpasses even the best AdamW results. We note that applying `Kappa-WS` initialization too late can cause instabilities due to weak regularization.

Since nnU-Net by default uses the SGD optimizer [48], we also test CPR to constrain optimization with the SGD optimizer in this context. As a more recent benchmark of segmentation performance, we report experiments on the Multi-Modality Abdominal Multi-Organ Segmentation Challenge 2022 [49]. This benchmark represents a very competitive segmentation challenge environment where differences as small as $0.1$ in Dice score can decide on challenge winners. As the experiments in Table J.1 suggest that on average 1k warm start steps, after the learning rate warmup leads to the best results, we resort to using 1k warm start steps for CPR since no learning rate warmup is present in the case of SGD in nnU-Net. As the weight decay value, we employ nnU-Net's default value of 3e-5. We show a strong performance out of the box in this context, improving on the very competitive nnU-Net baseline (89.45 Dice score) by a margin of 0.13 Dice points to a Dice score of 89.59. We note that hyperparameter tuning would most likely yield further performance improvements in this regard.

Table J.1: Results of medical image segmentation training on the BTCV, Heart, and BraTS datasets. We show the mean Dice score across 5 folds (3 for BTCV) for a range of weight decay values ($\gamma$) for AdamW and different warm start steps $s$ for CPR. The learning rate warmup is $2k$.

|  | SGD | | | | | SGD+CPR | | | |
| --- | --- | --- | --- | --- | --- | --- | --- | --- | --- |
|  | 1e-5 | 1e-4 | 1e-3 | 1e-2 | 1e-1 | 1k | 2k | 3k | 4k |
| BTCV | 83.04 | 83.1 | 83.17 | 83.99 | 73.92 | 81.17 | 84.14 | **84.23** | 55.41 |
| Heart | 92.92 | 92.75 | 92.88 | 92.9 | 92.85 | 92.77 | **93.18** | 93.16 | 74.44 |
| BraTS | 75.85 | 76.01 | 76.22 | 76.12 | 75.42 | 75.29 | 76.46 | **76.65** | 75.63 |

Table J.2: Hyperparameters of the medical image segmentation experiments.

| Parameter | Value |
| --- | --- |
| Fold | 0,1,2,3,4 |
| Dataset | BTCV, Heart, BraTS |
| Preprocessing | Default nnU-Net preprocessing [40] |
| Batch size | 2 (following [40] |
| Patch size | (48x192x192) BTCV, (80x192x160) Heart, (128x128x128) BraTS |
| Training Steps | 125k (BTCV), 25k (Heart &BraTS) |
| Model | 3d fullres U-Net (following [40]) |
| Optimizer | AdamW / AdamCPR |
| Learning Rate | 0.01 |
| Beta1 | 0.9 |
| Beta2 | 0.99 |
| Weight Decay | $1e-5\ldots1e-1$ (AdamW) |
| Lr Schedule | Polynomial decay with warmup |
| Lr Warmup Steps | 2000 |
| Lr Polynomial exponent | 0.9 |
| CPR-$\mu$ | 1.0 |
| CPR-$\kappa$ | 1.0 |
| CPR-$k$ | False |
| CPR-$\kappa$ warm-start steps | $1000\ldots4000$ |
| Adaptive Bounds | False |

# K Experiments on Fine-tuning a Large Language Model

Table K.1: Hyperparameters for the fine-tuning an LLM experiment.

| Parameter | Value |
|---|:---:|
| Model Name | mistralai/Mistral-7B-Instruct-v0.2 |
| Replace Layer | q_proj, v_proj, k_proj, o_proj, gate_proj, up_proj, down_proj |
| Learning Rate | 0.0005 |
| Warmup Steps | 50 |
| Pubmedqa Artificial Samples | 100000 |
| Epochs | 1 |
| Batch Size | 128 |
| Lora R | 128 |
| Lora Alpha | 1.0 |

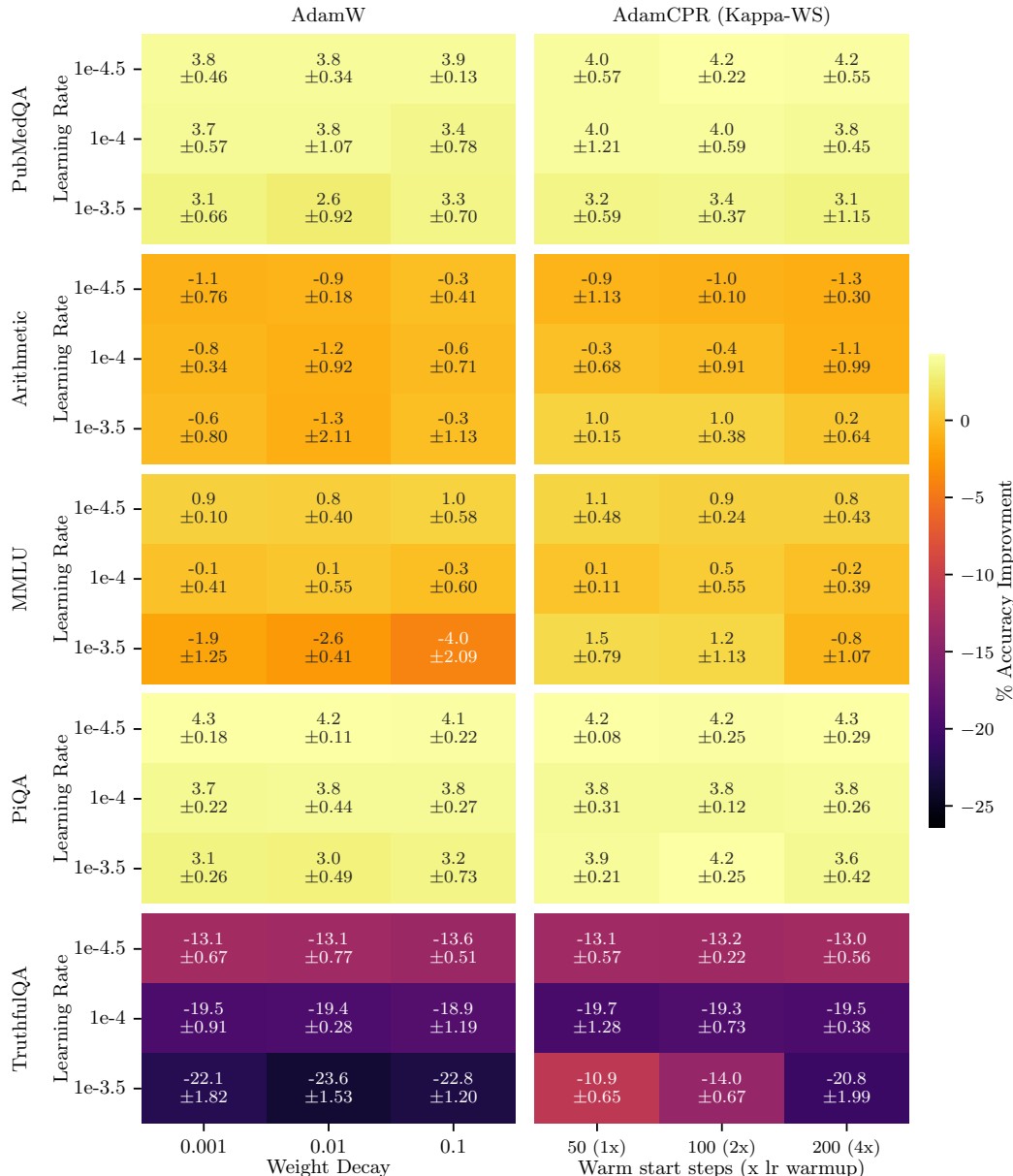

Figure K.1: Percentage of performance change before and after fineuning Mistral 7B with pubmedQA artificial data with the use of AdamW (left) and AdamCPR (right). AdamCPR uses the $L_2$ norm as a regularization function and `Kappa-WS`. We use a learning rate warm-up of 50 steps. The heatmap shows the mean performance and standard deviation across three random seeds. We use the *Arithmetic* dataset with 10 tests that involve simple arithmetic problems in natural language [50], the comprehensive *MMLU* benchmark [51], the *PiQA* benchmark on reasoning about physical commonsense in natural language [52], and the *TruthfulQA* benchmark, which evaluates models' abilities to mimic human falsehoods [39].

