# OpenReview forum: "Improving Deep Learning Optimization through Constrained Parameter Regularization"
_NeurIPS.cc/2024/Conference — NeurIPS 2024 poster_

### Official Review · Reviewer_dfuS · 2024-07-08

**Soundness:** 3
**Presentation:** 3
**Contribution:** 3
**Rating:** 7
**Confidence:** 3

**Summary:**

The authors propose a form of regularization which adjusts the regularization strength based on the weights being inside or outside of a certain norm bound. A violation of the bound results in an increasing penalty and coefficient, while conforming to the bound results in a decreasing or zero regularization penalty.

**Strengths:**

- The method proposes a way to adaptively control the strength of regularization for weigth groups within a model
- The model intuitively makes sense, as the regularization term is meant to constrain weights to within a certain region. However, current optimizers leave the regularization in place regardless of how well the weights are conforming to it.

**Weaknesses:**

- The statement on L123-124 is hard to accept, and I think it needs more explanation. Why if $F(x)$ not suitable for gradient based optimization? It seems that if $F$ and $c$ are both differentiable, then it indeed would provide iseful information to restore the feasibility of an infeasible $x$. Am I missing something here?

- It would be interesting to know what effect the training has on the rubustness to OOD inputs or noise perturbed inputs. Is it possible to run some tests on the trained models related to CIFAR100-C and AdvSQUAD or AdvGLUE (or other suitable language dataset to measure robustness)? I ask this because I suspect that the smaller weight norm maintained throughout training would have the effect of not being overly reliant on particular weights in the model, which would likely result in a model which is more robust to OOD and adversarial inputs.

- The minor runtime overhead mentioned in the paper is actually on the order of 5-6%. The definition of minor is subjective, and I would not consider a 5% increase in runtime minor. I think it would be better to state the actual runtime increase within the main text of the paper to give the reader a better idea of the cost.

## Minor

- L70: pertaining --> pretraining

**Questions:**

Can you add the OOD or adversarial tests mentioned above?

**Limitations:**

The limitations are discussed

---

> ### Author Rebuttal · Authors · 2024-08-07
>
> Thank you for your thoughtful review and valuable feedback. We appreciate your positive assessment of our method's intuition and potential for adaptive regularization. We have carefully considered your comments and would like to address them as follows:
>
> > The statement on L123-124 is hard to accept, and I think it needs more explanation. Why if 𝐹(𝑥) not suitable for gradient based optimization? It seems that if 𝐹 and 𝑐 are both differentiable, then it indeed would provide iseful information to restore the feasibility of an infeasible 𝑥. Am I missing something here?
>
> Even if $f(x)$ and $c(x)$ are differentiable, $F(x)$ is not differentiable for $c(x) > 0$. This is because of the maximization over $\lambda$ in F(x).
>
> $\underset{x}{\operatorname{minimize}} \; F(x) \quad \text{with} \quad F(x) = \max_{\lambda \ge 0} \; f(x) + \lambda \cdot c(x)$.
>
> The function F(x) (that is to minimize with respect to x) has an inner maximization with respect to $\lambda$. If $c(x) > 0$, in other words, if x is infeasible, $\lambda \to \infty$ “maximizes” $F(x)$ with respect to $\lambda$. Hence, for $c(x) >0, F(x)$ jumps to $\infty$ due to the maximization over \lambda. On the other hand, for $c(x) < 0$, when x is feasible, the maximization over \lambda yields $\lambda = 0$.
> We can thus alternatively write $F(x)$ as
> $F(x) = f(x)   if c(x) >= 0$     and    $F(x) =  \infty   if c(x) < 0$
> From this formulation, it is evident that we cannot run gradient based optimization on this objective. As soon as we encounter an infeasible x, we receive an input of $\infty$.
> The smoothed approximation $\hat{F}$ in Equation 1 addresses this effect by adding a quadratic term for $\lambda$ which prevents the maximization from returning $\infty$. Does this clarify our statement in L123-124?
> However, we agree that this point was not well-articulated. We will revise this section to provide a more precise and accurate explanation of the optimization challenges and how our approach addresses them.
>
>
> > It would be interesting to know what effect the training has on the rubustness to OOD inputs or noise perturbed inputs. Is it possible to run some tests on the trained models related to CIFAR100-C and AdvSQUAD or AdvGLUE (or other suitable language dataset to measure robustness)? I ask this because I suspect that the smaller weight norm maintained throughout training would have the effect of not being overly reliant on particular weights in the model, which would likely result in a model which is more robust to OOD and adversarial inputs. Can you add the OOD or adversarial tests mentioned above?
>
> We performed an additional experiment in CIFAR100-C, see the general response and Rebuttal PDF Figure 2. We found that AdamCPR outperforms AdamW which could indicate that CPR leads to a more robust optimization. We see that AdamCPR performs better than AdamW with Kappa-WS but not with Kappa-IP. None of the optimizer and hyperparameter configurations lead to an outstanding performance on this task, we wouldn’t claim that CPR is particularly good for noisy data. However, a qualified analysis of the robustness will have the scope of a separate paper and could be an interesting follow-up work.
>
>
> > The minor runtime overhead mentioned in the paper is actually on the order of 5-6%. The definition of minor is subjective, and I would not consider a 5% increase in runtime minor. I think it would be better to state the actual runtime increase within the main text of the paper to give the reader a better idea of the cost.
>
>
> We appreciate your perspective on the runtime increase. Upon reflection, we agree that characterizing a 5-6% increase as "minor" may be subjective. In the revised version, we will explicitly state the runtime increase in the main text.
>
> Minor correction: Thank you for catching the typo on L70. We will correct "pertaining" to "pretraining" in the revised version.
>
> Thanks again for your review and useful thoughts. Might we kindly ask you to increase your score in case we address your named concerns?

---

> > ### Comment · Reviewer_dfuS · 2024-08-12
> > **Thank you for the responses.**
> >
> > Thank you for the responses. The authors have adequately answered my questions. As I was already quite positive of the work, I will maintain my current score.

---

### Official Review · Reviewer_4Jjm · 2024-07-11

**Soundness:** 3
**Presentation:** 3
**Contribution:** 2
**Rating:** 6
**Confidence:** 2

**Summary:**

This paper presents Constrained Parameter Regularization (CPR) as an alternative to traditional weight decay.
CPR enforces an upper bound on the L2-norm of individual parameter matrices. It frames learning as a constraint optimization problem solved with the augmented Lagrangian method and can be integrated seamlessly with gradient-based optimizers.
During training, CPR dynamically tailors regularization and reduces the need for hyperparameter selection.
Empirical results on computer vision and language modeling tasks demonstrate its effectiveness.

**Strengths:**

- The proposed CPR is straightforward to implement and can be integrated with existing gradient-based optimization methods.
- Experiments demonstrate CPR’s effectiveness in various tasks compared to traditional weight decay. Results also show that the performance of CPR is robust to hyperparameter selection and needs less training budget.
- The paper is well-written and easy to understand, making the concepts accessible.

**Weaknesses:**

The general idea of using adaptive regularization does not seem very novel. However, the reviewer does not specialize in this area and won't view this as a significant flaw given CPR’s experimental results.

**Questions:**

1. CPR constrains parameters with an L2 norm upper bound. Will this method impede the model’s learning ability under complex tasks, e.g., very slow convergence or loss of training accuracy?
2. It would be interesting to investigate CPR’s robustness to noise and distributional shifts. How does CPR perform when the training data is noisy or comes from a different distribution than the test data?

---

> ### Author Rebuttal · Authors · 2024-08-07
>
> Thank you for the thoughtful review. We appreciate your positive comments on CPR's straightforward implementation, effectiveness across tasks, and clear presentation. We'd like to address your questions and concerns:
>
> > The general idea of using adaptive regularization does not seem very novel. However, the reviewer does not specialize in this area and won't view this as a significant flaw given CPR’s experimental results.
>
> We agree that the idea of adaptive regularization is not new, however, the idea of applying an upper bound to the $L_2$ norm (or any other statistical measurement) and enforcing this bound with the use of an augmented Lagrangian is novel.
> Furthermore, it gives access to a different hyperparameter ($\kappa$) for which more suitable initialization heuristics, see Kappa-WS/IP, can be developed than for AdamW’s static weight decay hyperparameter. CPR with the Kappa IP initialization is a hyperparameter-free regularization and the usage of the inflection point of the $L_2$ norm is novel too. Thereby, CPR outperforms AdamW on training with 350M parameter LLM experiments as well as ImageNet vision transformer training. No other regularization alternative to weight decay showed such successful and extensive evaluation.
>
> > CPR constrains parameters with an L2 norm upper bound. Will this method impede the model’s learning ability under complex tasks, e.g., very slow convergence or loss of training accuracy?
>
> Similar to weight decay, CPR limits the effective capacity of the model. This naturally interferes with training loss minimization. Nevertheless, regularization, in the form of constraints or penalties (such as weight decay) is usually required to avoid overfitting and increase generalization.
> We trained multiple complex tasks in our experiments, such as language models (GPT2s/m on OpenWebText) and image classification (ViT on ImageNet) and CPR surpassed traditional weight decay.
>
>
> > It would be interesting to investigate CPR’s robustness to noise and distributional shifts. How does CPR perform when the training data is noisy or comes from a different distribution than the test data?
>
> To test the performance of CPR on noisy data, we performed an additional experiment with the use of the noisy CIFAR100-C dataset [1] as mentioned in the general response. The results can be found in the Rebuttal PDF Figure 2. We see that AdamCPR outperforms AdamW which could indicate that CPR leads to a more robust optimization. We see that AdamCPR performs better than AdamW with Kappa-WS but not with Kappa-IP. None of the optimizer and hyperparameter configurations lead to an outstanding performance on this task, we wouldn’t claim that CPR is particularly good for noisy data. However, a qualified analysis of the robustness will have the scope of a separate paper and could be an interesting follow-up work.
>
> We appreciate your openness about your expertise, your fair assessment of CPR's contributions, and your questions. We believe that we have addressed them and that adding the results on noisy data (and ImageNet) will make the paper stronger. If you agree, might we kindly ask you to increase your score? We would be happy to provide any additional information or clarifications if needed.
>
> ___
>
> [1] Hendrycks, Dan, and Thomas Dietterich. "Benchmarking Neural Network Robustness to Common Corruptions and Perturbations." International Conference on Learning Representations. 2018.

---

> > ### Comment · Reviewer_4Jjm · 2024-08-12
> > **Official Comment by Reviewer 4Jjm**
> >
> > Thanks for the detailed response. I have no further questions and I have raised my score to 6.

---

### Official Review · Reviewer_M5LW · 2024-07-12

**Soundness:** 3
**Presentation:** 3
**Contribution:** 3
**Rating:** 7
**Confidence:** 2

**Summary:**

This paper illustrates a new training algorithm to improve the weight decay strategy. Instead of giving the same strength to all weights in weight decay, the proposed method penalizes only the elements that are larger than the threshold. Based on extensive evaluation, the proposed method performs better than AdamW/SGD+weight decay.

**Strengths:**

1. The experiment is strong enough to support the proposed method.
2. The intuition is easy to understand; that is, instead of using the same weight decay strength for all weights, the strength should be different for different weights since the importance is not the same.

**Weaknesses:**

1. As mentioned in the paper, the computational cost is higher.
2. The evaluation of SGD can be improved since it is well known that the best test accuracy can usually be achieved when using SGD with weight decay on image classification tasks.

**Questions:**

See weakness.

---

> ### Author Rebuttal · Authors · 2024-08-07
>
> Thank you for your thoughtful review and feedback on our paper. We appreciate your positive assessment of the strength of our experiments and the intuitive nature of our method.
> We would like to address the weaknesses you identified as follows:
> > As mentioned in the paper, the computational cost is higher.
>
> We acknowledge that there is a small computational overhead with our method depending on the model size as we analyze in Appendix H “Runtime Analysis on LLM training” of the original submission. However, the benefits in performance clearly outweigh this modest increase in cost as detailed in the following.
> For smaller models and larger batch sizes, the overhead is negligible (less than 1%). Even for our largest tested model (GPT2-XL with 1.5B parameters), the overhead was only 5.76% with a batch size of 1 and decreased to 2.44% at the maximum possible batch size. Importantly, this small increase in compute time translates to significant improvements in model performance and reduced total training time. For example, with GPT2s, we achieved the same performance as AdamW in only 2/3 of the training budget (Figure 1).
>
>
> > The evaluation of SGD can be improved since it is well known that the best test accuracy can usually be achieved when using SGD with weight decay on image classification tasks.
>
> We thank you for this point and, as mentioned in the general response, we performed additional experiments with SGD and SGD with CPR on the CIFAR100/ResNet18 task. We trained each configuration three times with random seeds and report the mean percentage of correct labels and standard deviation of the experiments. The results are in the Rebuttal PDF in Figure 1. We find that SGD with CPR outperforms SGD with weight decay when using the Kappa-WS initialization. However, the IP initialization seems not to work with the use of SGD, probably due to the changed convergence behavior in contrast to Adam. We will add this experiment and findings to our paper.
>
> Thanks again for your constructive feedback and the opportunity to improve our work. Might we kindly ask you to increase your score in case we address your concerns? We would be happy to provide any additional information or clarifications if needed.

---

> > ### Comment · Reviewer_M5LW · 2024-08-10
> >
> > Thank you for your response! I don't have other concerns so I keep my score.

---

### Official Review · Reviewer_TAzm · 2024-07-12

**Soundness:** 3
**Presentation:** 3
**Contribution:** 2
**Rating:** 4
**Confidence:** 3

**Summary:**

The paper introduces Constrained Parameter Regularization (CPR), a regularization technique for deep learning by dynamically tailoring regularization to individual parameters. CPR sets upper bounds on statistical measures of parameter matrices and reduces the learning into a constraint optimization problem. The authors conduct a series of experiments to evaluate the performance of the proposed method.

**Strengths:**

- The article provides an accurate formal description of the background and derivation process of the algorithm, which is clear and straightforward, making it easy to understand.

- The article theoretically explains the differences and connections between the CPR algorithm and weight decay algorithms, providing a good summary of existing optimization methods. It designed multiple initialization methods for the constraint upper bound κ, and extensively verified its effectiveness through experiments in various deep learning domains.

- The experimental results verify that the proposed method is techinically correct.

**Weaknesses:**

- The optimization algorithm lacks theoretical analysis on aspects such as convergence, which needs to be further analyzed using more rigorous mathematical language. Additionally, the initialization algorithm designed lacks theoretical support.

- The authors conduct experiments on CIFAR100, which is too small to show the superiority of a training method for deep learning. The authors need to give the results on ImageNET or larger-sized datasets.

- The proposed methods have more learnable parameters $\lambda$ and $\kappa$. The authoes need to report the addtional computational and memory costs.

**Questions:**

Please refer to the comments on weaknesses.

**Limitations:**

The authors have adequately addressed the limitations

---

> ### Author Rebuttal · Authors · 2024-08-07
>
> We thank you for your time and effort in reviewing our paper. We appreciate your positive feedback on the clarity of our presentation and the extensive empirical validation across multiple deep learning domains. In the following we carefully considered your concerns and questions:
>
> > The optimization algorithm lacks theoretical analysis on aspects such as convergence, which needs to be further analyzed using more rigorous mathematical language.
>
> We acknowledge this limitation and agree that a rigorous theoretical analysis would strengthen the method. While we focused primarily on empirical validation in this initial work, we recognize the importance of theoretical foundations.
> However, a theoretical analysis of CPR exceeds the scope of this work. In particular, providing a fair analysis of the interplay between momentum methods and the decoupled updates (in the spirit of weight decay) likely requires a different treatment than the analysis of the standard augmented lagrangian methods.
> What is more, the incomplete optimization (single update step) of the loss, before $\lambda$ is updated, further complicates the analysis as we cannot rely on the classic interpretation of the update of lambda as a step of dual gradient ascent. Also, since the update of the parameters with $\nabla R(x) * \lambda$ is not multiplied with the learning rate, we cannot expect lambda to converge while the learning rate is still decreasing, as stationarity is only reached when the update direction of the optimizer and $\lambda * R(\theta)$ cancel.
>
> Therefore, we focused in this work on the empirical evaluation of the method to provide deep learning practitioners with a powerful alternative to weight decay and in the case of Kappa-IP, without the need for tuning a regularization hyperparameter (like the weight decay $\gamma$).
>
>
> > Additionally, the initialization algorithm designed lacks theoretical support.
>
> Both, gamma in weight decay and kappa in CPR are hyperparameters to be determined experimentally.
> This is because any measure of model complexity can only be estimated through the validation data. Without this, the correct model complexity, measured, e.g., through a regularization function, can only be assumed.
> Such assumptions are implicitly made through the choice of the regularization parameters.
> For this choice, we propose an initialization heuristic (Kappa-IP) and show that it works well across different tasks, datasets, and architectures. From this, we conclude based on experimental evidence, that the assumption underlying it generalizes well. We provide empirical evidence for the Kappa-IP initialization in experiments on LLMs (Paper Figure 3, Figure 5) and ImageNet (Rebuttal PDF Table 1).
>
> > The authors conduct experiments on CIFAR100, which is too small to show the superiority of a training method for deep learning. The authors need to give the results on ImageNET or larger-sized datasets.
>
>
> We performed ImageNet pretraining experiments on a vision transformer (DeiT [1]) with two sizes, small with 22M parameters and base with 86M. Unfortunately, we only got the small experiments done until the rebuttal deadline but will provide the base results in a comment on the weekend. The results can be found in Table 1 in the Rebuttal PDF.
> In contrast to [1], we found that a higher weight decay works better. However, AdamCPR outperforms AdamW with both kappa initializations, a tuned warm start kappa initialization (Kappa-WS), and the hyperparameter-free Kappa-IP. In the small model training, we outperform weight decay by 0.84% without the need for tuning a regularization hyperparameter. We measured a very minor runtime increase of 0.02% when using CPR in comparison to AdamW.
>
> We also want to stress that the GPT2s (124M parameter) and GPT2m (345M) are trained on the OpenWebText dataset with ~9B token, which is not small and a common experimental setting in LLM-pretraining.
>
>
> > The proposed methods have more learnable parameters 𝜆 and 𝜅. The authoes need to report the addtional computational and memory costs.
>
> Note that \lambda and \kappa are only scalars (one per parameter group / weight matrix). These parameters are not learnable but are used by the Lagrangian optimization. Their updates do not require gradient computations but only the computation of $R(\theta)$.
>
> For the computational costs, we would like to refer you to Appendix H “Runtime Analysis on LLM training” of the original submission. We have conducted a detailed analysis of the computational overhead introduced by CPR. For GPT2-small (124M params), CPR introduces only a 0.4% runtime increase compared to AdamW. For GPT2-medium (354M params), the overhead is 2.4%. Importantly, this small increase in compute time translates to significant improvements in model performance and reduced total training time. For example, with GPT2s, we achieved the same performance as AdamW while requiring only 2/3 of the training budget (Figure 1).
>
>
> We thank you again for your constructive feedback and the opportunity to improve our work. Might we kindly ask you to reevaluate your scoring and increase the score in case we address your concerns? If any concerns remain we are more than happy to discuss these via comments in OpenReview.
> ___
>
> [1] Touvron, Hugo, et al. "Training data-efficient image transformers & distillation through attention." International conference on machine learning. PMLR, 2021.

---

> > ### Comment · Reviewer_TAzm · 2024-08-12
> >
> > Thanks for the authors' response.
> >
> > Most of mu concerns have been addressed. But as the poposed  method is actually a training algorithm/technique for deep neural networks, I think it would be better and necessary to provide some theoretical analysis for it. I increased my rating to 4.

---

### Official Review · Reviewer_gmoY · 2024-07-17

**Soundness:** 3
**Presentation:** 4
**Contribution:** 3
**Rating:** 5
**Confidence:** 4

**Summary:**

The paper proposes a new regularization technique, Constrained Parameter Regression (CPR), to replace weight decay for training deep learning models. Conventional weight decay penalizes significant weight deviation. However, this can be too restrictive since some layers may need a larger deviation. Instead of applying the same regularization strength for all layers as in weight decay, the paper uses different hyperparameters for each layer, resulting in a more flexible regularization strength. Intuitively, a model with a larger weight $||\theta||$ is allowed more deviation under CPR. The method is theoretically motivated by the augmented Lagrangian method and tested on multiple tasks and benchmarks.

**Strengths:**

* The method is principledly motivated and intuitively explained. Overall, the paper is very well-written.

* The paper proposes multiple alternatives for setting the method's hyper-parameters, taking into account efficiency and tuning flexibility.

* The paper demonstrates superior performance on multiple tasks with modern deep learning models, including large language models.

**Weaknesses:**

* **The regularization strength (upper bound $k$) relies on empirical observations and intuition**. While the overall method is theoretically motivated, the upper bound $k$ setting relies mainly on intuition. Specifically, the paper does not explain why $k\leftarrow R(\theta)$ is a good strategy. Intuitively, this strategy indicates that a larger weight $||\theta||$ should have less regularization.

* **Computer vision tasks are limited**. For classification, computer vision experiments are conducted on relatively small-scale datasets, such as CIFAR100. The paper should consider using moderate-to-large-scale datasets with well-established benchmarks, such as ImageNet, to make the empirical evidence more convincing. For segmentation, popular benchmarks, such as MSCOCO, could be a better choice to establish the superiority of this regularization.

**Questions:**

* Why does the inflection point of the regularization function $\Delta\Delta R$ reflect a saturation of performance? Shouldn't the inflection point of the loss function $L$ be a better fit for this?

* Could the authors report results on ImageNet? If time is constrained, fine-tuning a pre-trained model, such as CLIP's pre-trained ViT-B [1], to ImageNet is a good choice.

* Is there any empirical evidence supporting the design choice $k\leftarrow R(\theta)$? It would be great if the authors could discuss this choice in detail.

* Prior works in hyper-optimization [2,3] show how to optimize hyper-parameters in optimizers differetiably. This could be a valid future direction for optimizing the upper bound $k$. A new work uses hyper-optimization in an optimizer for fine-tuning [4], which shares a similar spirit to this paper at a high level.

[1] Radford, Alec, et al. "Learning transferable visual models from natural language supervision." International conference on machine learning. PMLR, 2021.

[2] Baydin, Atilim Gunes, et al. "Online learning rate adaptation with hypergradient descent." arXiv preprint arXiv:1703.04782 (2017).

[3] Chandra, Kartik, et al. "Gradient descent: The ultimate optimizer." Advances in Neural Information Processing Systems 35 (2022): 8214-8225.

[4] Tian, Junjiao, et al. "Fast trainable projection for robust fine-tuning." Advances in Neural Information Processing Systems 36 (2023).

**Limitations:**

There is no potential negative societal impact.

---

> ### Author Rebuttal · Authors · 2024-08-07
>
> Thank you for your thoughtful review of our paper on Constrained Parameter Regularization (CPR). We appreciate your positive feedback on the method's principled motivation, clear presentation, and strong empirical results across multiple tasks.
>
> Regarding your summary, we would like to point out that we do not use “different hyperparameters for each layer”. We introduce only one hyperparameter for the regularization, namely the kappa initialization (or without any hyperparameter when using Kappa IP) but our method regularizes each layer individually with an additional scalar variable for each layer. We also would disagree that “a model with a larger weight $||𝜃||$ is allowed more deviation under CPR” but CPR enforces a norm of $||𝜃|| \le \kappa$ earlier in model training.
>
> In the following, we address your questions:
>
> > Why does the inflection point of the regularization function ΔΔ𝑅 reflect a saturation of performance? Shouldn't the inflection point of the loss function 𝐿 be a better fit for this?
>
> The inflection point tries to identify a point at which increasing the model complexity (measured through $R$) starts becoming less relevant for reducing the loss than in previous iterations. In other words, the benefit for increasing the model complexity further, starts saturating. Hence, we try to capture the point where $\Delta R$ (the change in $R$ over successive iterations) starts to decrease. Also, note that each parameter group (weight matrix) has its own inflection point.
>
> > Could the authors report results on ImageNet? If time is constrained, fine-tuning a pre-trained model, such as CLIP's pre-trained ViT-B [1], to ImageNet is a good choice.
>
> We performed both, new experiments on pretraining a vision transformer on ImageNet as well as finetuning CLIP, as mentioned in the general response.
> We performed ImageNet pretraining experiments on a vision transformer (DeiT [1]) with two sizes, small with 22M parameters and base with 86M. Unfortunately, we only got the small experiments done until the rebuttal deadline but will provide the base results in a comment on the weekend. The results can be found in Table 1 in the Rebuttal PDF.
> In contrast to [1], we found that a higher weight decay works better. However, AdamCPR outperforms AdamW with both kappa initializations, a tuned warm start kappa initialization (Kappa-WS), and the hyperparameter-free Kappa-IP. In the small model training, we outperform weight decay by 0.84% without the need for tuning a regularization hyperparameter.  We measured a very minor runtime increase of 0.02% when using CPR in comparison to AdamW.
> Additionally, we performed finetuning experiments, as you mentioned, with a CLIP ViT-B to ImagenNet. The results can be found in Table 2 in the Rebuttal PDF.  AdamCPR outperforms AdamW with the use of a tuned Kappa-WS initialization and the Kappa-IP initialization is on par with the best weight decay hyperparametrization.
>
> > Is there any empirical evidence supporting the design choice 𝑘←𝑅(𝜃)? It would be great if the authors could discuss this choice in detail.
>
> Note that $\kappa$ is a bound for the value of $R(\theta)$ (Section 4.3). Choosing $\kappa \gets R(\theta)$ for a specific $R(\theta)$ observed in the training run (as done in Kappa-WS and Kappa-IP), ensures that the bound is active, while also not restricting the training so much that it fails to reduce the loss. In particular, it is ensured that when the bound becomes active, a value of $R(\theta)$ is enforced for which healthy training dynamics are expected.
>
> We also performed an empirical evaluation of the different initialization methods which can be found in Appendix E, Figure E.1. While Kappa-kI$_0$ (kappa initialization depending on the initial $R(\theta)$) also leads to good performance, we found Kappa-WS ( $\kappa \gets R(\theta)$ after $x$ warm start steps) to perform better over a larger span of different learning rates.
>
>
> > Prior works in hyper-optimization show how to optimize hyper-parameters in optimizers differentiable. This could be a valid future direction for optimizing the upper bound 𝑘. A new work uses hyper-optimization in an optimizer for fine-tuning, which shares a similar spirit to this paper at a high level.
>
> We thank you for this suggestion. Optimizing for $\kappa$ could be an interesting direction. One way to get gradients for $\kappa$ could be to use implicit differentiation using the KKT conditions of the CPR problem (see, e.g. [2]). However, this would likely not be straightforward, and while this is an interesting avenue for future extensions of this work, we hope to have demonstrated in the paper that our heuristics, in particular choosing the inflection point for setting $\kappa$, already provide strong results for negligible computational overhead.
>
> Thanks again for your review and useful thoughts.  We believe that we have addressed your concerns and the additional positive results on ImageNet training strengthen our paper. Might we kindly ask you to increase your score in case we address your named concerns?  We would be happy to provide any additional information or clarifications if needed.
>
> ____
>
> [1] Touvron, Hugo, et al. "Training data-efficient image transformers & distillation through attention." International conference on machine learning. PMLR, 2021.
>
> [2]  Blondel, M., “Efficient and Modular Implicit Differentiation”, 2021. doi:10.48550/arXiv.2105.15183.

---

> > ### Comment · Reviewer_gmoY · 2024-08-11
> >
> > Thank the authors for their new experiments, and thank you for clarifying my misunderstandings of the paper.
> >
> > * I want to mention that ''additional scalar variables for each layer'' is what I meant by ''different hyperparameters for each layer''.
> >
> > * In the summary, I summarized the method as  " a larger weight is allowed more deviation". This is my logic. The goal is to keep $\||\theta\|| < \kappa$ (line 160), where $\kappa\propto R(\theta)$ (sec.4.3) and $R(\theta) = 1/2\||\theta\||^2_2$ (line 107). In other words, a weight with larger norm $\||\theta\||^2_2$ will have smaller regularization because $\kappa$ is large (a larger upper bound). Hence, it is allowed more deviation. Please let me know where I misunderstood the method. Thanks.

---

> > > ### Author Response · Authors · 2024-08-12
> > >
> > > We thank the reviewer for their response and clarification.
> > >  -  When we read “hyperparameters”, we think of some parameters to tune (beforehand). Since this is not the case we criticized it. We just misunderstood the wording.
> > >  - In principle, the degree of allowed deviation in our method corresponds to the value of $\kappa$ and $\kappa$ depends on the $\kappa$ initialization method. For example, if one uses the $\kappa$ initialization method that has fixed $\kappa$ for all weight matrices (Kappa-K) then it does not depend on the weights $\theta$ and $\kappa \not\propto R(\theta_t)$. If one uses a $\kappa$ initialization method which is dependent on $R(\theta_t)$, like Kappa-WS, then $\kappa \propto R(\theta_t)$ and we agree that the entries of weight matrices with larger $R(\theta_t)$ are allowed more deviation. Or in other words, a model with a larger weight $||\theta||$ is allowed more deviation under CPR when initializing $\kappa$ with a $\kappa$ initialization method which is $\kappa \propto R(\theta_t)$.

---

### Author Rebuttal · Authors · 2024-08-07

Dear Reviewer,

Thank you for your thorough and constructive reviews of our paper on Constrained Parameter Regularization (CPR). We greatly appreciate your thoughtful comments and the opportunity to address your concerns.

We have prepared a 1-page PDF with additional experimental results that directly address several of the points you raised:

- **ImageNet pre-training**: We compare AdamW and AdamCPR in a vision transformer [1] pertaining on ImageNet. We train a small DeiT[1] model with 22M parameters and a base model with 86M parameters with the use of the PyTorch Image Models library [2] for 300 epochs and with the configuration for the DeiT Paper but also with a 10x and 0.1x weight decay value.
Unfortunately, we only managed to run the smaller model experiments within the rebuttal week. The base model experiments are still running and we will announce the results in a comment on the weekend.
As seen in Table 1 of the Rebuttal PDF AdamCPR outperforms AdamW in the small DeiT training with both kappa initialization methods. Especially the hyperparameter free regularization with Kappa-IP performs best by outperforming the best AdamW run by 0.86%. In the case of this small model, we measured a very minor runtime increase of 0.02% when using CPR in comparison to AdamW (14.85h for AdamW and 14.89h for AdamCPR on 4xA100).

- **ImageNet finetuning**:  We appreciate the reviewer’s suggestion regarding the evaluation on ImageNet. Following the reviewer's recommendation, we conducted fine-tuning of CLIP’s ViT-B/32 model on the ImageNet dataset. We used the ViT-B/32 model pre-trained by CLIP. The model was fine-tuned for 10 epochs following the hyperparameter choices in [3], with the exception of the special classification head initialization. As in [3], we employ a learning rate of $3 \times 10^{-5}$, default PyTorch AdamW hyperparameters $\beta_1=0.9$, $\beta_2=0.999$ $\epsilon=10^{-8}$ and a cosine-annealing learning rate schedule with 500 warm-up steps. Due to time and compute constraints, the training was performed on a single GPU with a batch size of 512, compared to the original setup of 8 GPUs with a batch size of 512 each. In Table 2 of the Rebuttal PDF, we compare AdamW with different weight decay values to the proposed AdamCPR in different configurations, where we report the top-1 accuracy after finetuning. From these results, we see that the Kappa-WS initialization also leads to better results in this finetuning setting, comparing favorably to traditional weight decay.

- **SGD CPR experiments**: As mentioned by Reviewer M5LW, the best accuracy can be achieved in image classification (probably on CNNs, on ViTs is Adam used) with SGD and weight decay, So we performed additional experiments with SGD and SGDCPR on the CIFAR100/ResNet18 task. We used SGD with Nesterov momentum of 0.9 and configured the training similarly to the CIFAR100 experiments described in the paper. The results can be found in Figure 1 in the Rebuttal PDF. We trained each configuration three times with random seeds and report the mean percentage of correct labels and standard deviation of the experiments. We find that SGD with CPR outperforms SGD with weight decay when using the Kappa-WS initialization. However, the IP initialization seems not to work with the use of SGD, probably due to the changed convergence behavior in contrast to Adam.

- **CIFAR100-C experiments**: To evaluate CPR's robustness according to data noise, we've included experiments on training a ResNet18 on the noisy CIFAR100-C dataset [4]. The training setup and configuration are similar to the CIFAR100 experiments described in the paper. The results are visualized in Figure 2 in the Rebuttal PDF. We see that AdamCPR performs better than AdamW with Kappa-WS but not with Kappa-IP. None of the optimizer and hyperparameter configurations lead to an outstanding performance on this task, we wouldn’t claim that CPR is particularly good for noisy data. We trained each configuration three times with random seeds and reported the mean percentage of correct labels and standard deviation of the experiments.

These additional experiments significantly strengthen our empirical evaluation and address points you raised. We believe they underscore CPR's effectiveness across a broader range of scenarios and dataset scales. We will add all experiments to our paper. We answer individual questions and address further concerns in each response below.

___

[1] Touvron, Hugo, et al. "Training data-efficient image transformers & distillation through attention." International conference on machine learning. PMLR, 2021.


[2] Wightman, Ross, "PyTorch Image Models.", github.com/rwightman/pytorch-image-models, GitHub repository 2019.


[3] Wortsman, Mitchell, et al. "Robust fine-tuning of zero-shot models." Proceedings of the IEEE/CVF conference on computer vision and pattern recognition. 2022.


[4] Hendrycks, Dan, and Thomas Dietterich. "Benchmarking Neural Network Robustness to Common Corruptions and Perturbations." International Conference on Learning Representations. 2018.

---

### Author Response · Authors · 2024-08-13
**Additional ViT/ImageNet results**

Dear Reviewer,

Please find below the additional DeiT[1]/ImageNet results on the DeiT base model that we promised in our rebuttal (since they were still running):

| ImageNet Pretraining | | AdamW | | | AdamCPR | | | |
|----------------------|-|-------|-------|-------|---------|---------|---------|-----------|
| | | weight decay | | | Kappa WS (x lr-warmup) | | | Kappa IP |
| | | 0.005 | 0.05¹ | 0.5 | 1x | 2x | 4x | |
| DeiT-Small (22M) | Top-1 Accuracy (%) | 76.97 | 79.03 | 79.16 | 79.81 | 79.33 | 78.04 | **79.84** |
| DeiT-Base (86M) | Top-1 Accuracy (%) | 76.19 | 78.59 | 80.56 | **81.19** | 79.61 | TBA | 80.95 |

Each experiment runs on 4 A100, the small model for ~15h, and the base model for ~48h. We used the Timm library [2] for all ImageNet experiments and used the configuration described in the DeiT paper:

`train.py --seed 0 --dataset wds/imagenet --aa rand-m9-mstd0.5-inc1 --mixup 0.8 --cutmix 1.0 --reprob 0.25  --model deit_base_patch16_224 --color-jitter 0.3 --train-interpolation bicubic --lr 1e-3 --warmup-lr 1e-6 --min-lr 1e-5 --warmup-epochs 5 --drop-path 0.1 --opt-eps 1e-8 --epochs 300 --opt adamw --weight-decay 0.05 --sched cosine -b 256 --amp --torchscript --pin-mem -j 8 --sched-on-updates`

Unfortunately, even though we directly used this configuration described in the DeiT paper, it did not achieve the same results for the base model as reported in the DeiT paper. We do not know the reason for this, but differences in PyTorch version, etc can make a difference. Also, in our own experiments, we found the optimal weight decay value for AdamW to be higher and not 0.05 as described in the DeiT paper. However, we used the same Timm configuration for CPR and see again that CPR with both kappa initialization, warm start and inflection point, outperforms weight decay. Thanks again for your review and useful thoughts.
______
[1] Touvron, Hugo, et al. "Training data-efficient image transformers \& distillation through attention." International conference on machine learning. PMLR, 2021.

[2] Wightman, Ross, "PyTorch Image Models.", github.com/rwightman/pytorch-image-models, GitHub repository 2019.

---

### Decision · Program_Chairs · 2024-09-25

**Decision:**

Accept (poster)

**Comment:**

The submission received mixed ratings after the post-rebuttal discussion period. The main concern appears to be a lack of theoretical analysis. However, while a theoretical analysis would make the submission more appealing, the empirical results appear to be sufficient to merit acceptance. The authors are advised to take into account the detailed reviews as well as the clarifications provided in the rebuttal while revising the paper.